# Cell divisions both challenge and refine tissue boundaries in the *Drosophila* embryo

Veronica Castle[1,2,3], Merdeka Miles[4], Rafael Perez-Vicente[2,3,5], Rodrigo Fernandez-Gonzalez[1,2,3,6],* and Gonca Erdemci-Tandogan[4,*]

## ABSTRACT

Tissue boundaries pattern embryos, suppress tumours and provide directional cues. Tissue boundaries are associated with supracellular cables formed by actin and the molecular motor non-muscle myosin II. Actomyosin cables generate tension that prevents cell mixing. Whether other cellular behaviours contribute to the formation of linear interfaces between cell populations remains unclear. In the *Drosophila* embryo, an actomyosin-based boundary separates the ectoderm from the mesectoderm, a group of neuronal and glial progenitors. Mathematical modelling predicted that cell divisions in the ectoderm challenge the mesectoderm-ectoderm (ME) boundary. Consistent with this, suppressing ectoderm cell divisions *in vivo* prevented cell mixing across the ME boundary when actomyosin-based tension was lost. Our mathematical model also predicted that cell divisions sharpen the ME boundary by reducing tension and increasing cell motility in the ectoderm. We found that inhibiting ectoderm divisions *in vivo* reduced boundary linearity. Using laser ablation and cell tracking, we demonstrated that cell divisions reduced junctional tension and increased cell movement in the ectoderm. Together, our results reveal that cell divisions facilitate cellular rearrangements to increase fluidity in a previously unreported mechanism for boundary refinement.

KEY WORDS: Epithelial morphogenesis, Boundary formation, Tissue fluidity, Quantitative microscopy, Mathematical modelling, Biophysics

## INTRODUCTION

Tissue boundaries are crucial for patterning and growth during embryonic development (Meinhardt, 1983). In adults, tissue boundaries can minimize tumour malignancy by limiting cell invasion (Astin et al., 2010). Boundaries between tissues must withstand challenges from cell movement, cell division or cell death

[1]Department of Cell and Systems Biology, University of Toronto, Toronto, ON, M5S 3G5, Canada. [2]Translational Biology and Engineering Program, Ted Rogers Centre for Heart Research, University of Toronto, Toronto, ON, M5G 1M1, Canada. [3]Institute of Biomedical Engineering, University of Toronto, Toronto, ON, M5S 3G9, Canada. [4]Department of Physics and Astronomy, University of Western Ontario, London, ON, N6A 3K7, Canada. [5]Division of Engineering Science, University of Toronto, Toronto, ON, M5S 1A4, Canada. [6]Developmental and Stem Cell Biology Program, The Hospital for Sick Children, Toronto, ON, M5G 1X8, Canada.

*Authors for correspondence (rodrigo.fernandez.gonzalez@utoronto.ca; gerdemci@uwo.ca)

R.F.-G., 0000-0003-0770-744X; G.E.-T., 0000-0001-6955-8692

(Monier et al., 2010); thus, boundaries are often associated with the generation of mechanical force to maintain the separation of distinct cell populations (Heisenberg and Bellaïche, 2013). Despite their importance, the mechanisms by which boundaries are maintained and refined are still not well understood.

Tissue boundaries separate cell populations to ensure proper tissue organization (Sharrock and Sanson, 2020; Wang and Dahmann, 2020). For example, in the *Drosophila* wing imaginal disc, two boundaries resist cell mixing due to proliferation: one boundary separates the anterior and posterior compartments of the disc (Sharrock and Sanson, 2020; Wang and Dahmann, 2020), and a second boundary separates the dorsal and ventral compartments (Garcia-Bellido et al., 1973; Morata and Lawrence, 1975). In the vertebrate hindbrain, boundaries between rhombomeres facilitate the differentiation of distinct regions (Lumsden and Krumlauf, 1996; Moens et al., 1998), and loss of rhombomere boundaries can cause abnormal development of the cranial and nasal bones (Twigg et al., 2004; Davy et al., 2006). Boundaries can also segregate cancerous cells from healthy tissues. In the mouse intestine, a boundary surrounds carcinomas, and metastasis can only occur if the boundary is disrupted (Cortina et al., 2007). Similarly, in the prostate, the boundary between epithelial and stromal compartments is usually lost during tumour invasion (Foty and Steinberg, 2004). Thus, boundaries are a key feature for the development and maintenance of normal tissue organization, and disruption of tissue boundaries is associated with disease (Major and Irvine, 2006; Landsberg et al., 2009; Monier et al., 2010; Calzolari et al., 2014; Yu et al., 2021).

Mechanical force is fundamental to the establishment and maintenance of tissue boundaries. Boundaries often display an enrichment of the molecular motor non-muscle myosin II and filamentous actin (F-actin), forming supracellular cables (connected by cell-cell adhesive structures) that span the length of the boundary (Wang et al., 2020). Actomyosin cables are present in the anterior-posterior and dorsal-ventral boundaries of the *Drosophila* wing disc (Major and Irvine, 2006; Landsberg et al., 2009; Umetsu et al., 2014), and also in the embryo, in the parasegmental boundaries between repeating developmental units (Monier et al., 2010), or around salivary gland precursors (Sanchez-Corrales et al., 2018). Beyond *Drosophila*, actomyosin cables characterize the boundaries between the notochord and presomitic mesoderm in *Xenopus* (Rohani et al., 2011), and around the eye field of zebrafish (Cavodeassi et al., 2013). Actomyosin cables at boundaries generate forces that smoothen the interface between adjacent tissues (Landsberg et al., 2009; Monier et al., 2010; Calzolari et al., 2014).

The mesectoderm in the *Drosophila* embryo recently emerged as a system with which to study tissue boundaries (Yu et al., 2021). The mesectoderm separates the mesoderm (ventral) from the ectoderm (lateral) on both sides of the ventral midline (Movie 1). At the end of mesoderm internalization, contralateral mesectoderm

cells meet at the midline and seal the mesoderm inside the embryo. Mesectoderm cells then undergo a single round of oriented divisions that facilitate axis elongation (Wang et al., 2017; Camuglia et al., 2022). As the embryo develops, the mesectoderm is internalized, giving rise to glia and neurons of the central nervous system (Jacobs and Goodman, 1989; Klämbt et al., 1991; Tepass and Hartenstein, 1994; Wheeler et al., 2006).

A tissue boundary separates the mesectoderm from the ectoderm (Yu et al., 2021). After dividing, mesectoderm cells reverse their planar polarity and localize both actin and myosin II at the interface with the ectoderm, forming supracellular cables flanking the mesectoderm. The supracellular cables sustain increased tension. Increased tension at the ME boundary prevents ectoderm invasion of the mesectodermal region, allowing the mesectoderm to internalize in a timely manner. Importantly, and in contrast to compartment boundaries in the wing disc or the embryo (Major and Irvine, 2005; Monier et al., 2010; Röper, 2012), the actomyosin cable at the ME boundary is disassembled over time, in a process thought to contribute to the internalization of the mesectoderm (Yu et al., 2021). Despite the reduction in myosin levels, the ME boundary remains linear. The mechanisms that maintain boundary linearity while myosin levels decrease are not understood.

Cell divisions challenge tissue boundaries. During division, cells round up through a reduction in adhesion to adjacent cells and an increase in osmotic pressure (Stewart et al., 2011; Fischer-Friedrich et al., 2014). Thus, dividing cells generate forces on adjacent cells. When cell divisions occur next to a boundary, they can transiently deform the boundary (Monier et al., 2010). Actomyosin contractility at the boundary generates tension in response to deformation, effectively pushing dividing cells into their original compartment and restoring the smooth interface between tissues. Notably, ectoderm cells divide next to the ME boundary (Wang et al., 2017). However, the impact of ectoderm cell divisions on the ME boundary has not been investigated.

## RESULTS
### Mathematical modelling predicts that cell divisions challenge the ME boundary

We used computational modelling to investigate if ectoderm cell divisions play a role in the dynamics of the ME boundary. To this end, we developed an adaptive vertex model with time-varying parameters. The model was initialized with cell geometries corresponding to an *in vivo* configuration after mesectoderm cells divide and before ectoderm divisions begin (Fig. 1A, Movie 2). Cells were assigned an energy that increased as cell area or perimeter deviated from a target value, with target values defined differently for ectoderm and mesectoderm cells based on *in vivo* measurements (Yu et al., 2021) (see Materials and Methods). To simulate the myosin cables at the ME interface, we incorporated a time-dependent line tension between ectoderm and mesectoderm cells that decreased over time, consistent with the disassembly of the myosin cable at the ME boundary. Ectoderm cells were stochastically selected to divide, with a frequency and orientation based on experimental data (Yu et al., 2021): cells adjacent to the ME boundary oriented their spindle preferentially along the dorsal-ventral axis, while the rest of the cells divided with random spindle orientations. Energy minimization guided the evolution of the cells in the model.

Our computational model predicted that the ME boundary prevents cell mixing. We found that acutely reducing tension at the ME boundary significantly increased the roughness of the ME interface by 36±1% (mean±s.e.m.) within 10 min ($P<0.001$), and by an additional 16±2% over the next 30 min, for a total 52±2%

increase by 40 min ($P<0.001$, Fig. 1A,B,E-G). Contralateral ectodermal cells came in close proximity or established premature contacts in the absence of ME boundaries (Fig. 1B, arrowhead). Thus, our model suggests that tension at the ME boundary maintains a smooth interface between ectoderm and mesectoderm, and prevents cell mixing.

To investigate if the ME boundary prevents cell mixing by resisting ectoderm cell divisions, we used the model to quantify boundary dynamics when ectoderm divisions were inhibited (Fig. 1C,D). Inhibiting ectodermal divisions did not prevent the increase in ME boundary roughness 10 min after losing tension at the boundary, but reduced the secondary increase in roughness between 10 and 40 min by 82% ($P<0.01$, Fig. 1E-G), and prevented formation of ectoderm bridges. Thus, modelling predicts that cell divisions provide a morphogenetic stress (not the only one) that challenges boundary linearity.

To further establish how ectoderm cell divisions may impact the ME boundary, we simulated scenarios altering the orientation of cell divisions adjacent to the boundary. We found that randomizing the angle of division of ectoderm cells adjacent to the boundary (Fig. S1A,B,D,E,G), or rotating the angle by 90°, such that the spindle was oriented preferentially along the anterior-posterior axis (Fig. S1A,C,D,F,G), did not affect the increase in roughness when tension at the boundary decreased, or the rescue of roughness when cell divisions were inhibited, consistent with the relatively small number of ectoderm cells adjacent to the boundary. Instead, the rescue in boundary roughness in the absence of tension could be accomplished only when all cell divisions (Fig. S2A,B,F) or cell divisions far from the boundary (Fig. S2C-F) were inhibited or reduced in number, suggesting that the number of ectoderm cell divisions, rather than their position or orientation, challenges the ME interface. Importantly, limiting the apical constriction of mesectoderm cells in the simulations did not affect the increase in ME boundary roughness when actomyosin contractility was inhibited (Fig. S3A-B,E), or the rescue of the roughness phenotype when cell divisions were blocked (Fig. S3C-E), suggesting that changes in mesectoderm cell morphology do not challenge the ME boundary.

### Cell divisions challenge the ME boundary in living embryos

To test the prediction that the ME boundary resists ectoderm divisions to prevent cell mixing, we looked for methods to acutely inhibit ectoderm divisions *in vivo*, while still allowing mesectoderm cells to divide and the ME boundary to form. Dinaciclib inhibits cyclin-dependent kinases 1, 2, 5 and 9 (Parry et al., 2010), and it has been used to inhibit cell division in the *Drosophila* embryo (Akhmanova et al., 2022). We found that treatment with 500 μM of dinaciclib at the end of mesectoderm divisions prevented subsequent divisions in the ectoderm (Movie 3). To disrupt the cable at the ME boundary, we treated embryos with 10 mM of Y-27632, a Rok inhibitor, as we did before (Yu et al., 2021) (Movie 3). Notably, inhibiting myosin with Y-27632 did not prevent cell division (Movie 4): we measured 1.0±0.2 cells that completed cytokinesis/min in a 136×136 μm² area of the ectoderm in control embryos, and 0.7±0.1 cells that completed cytokinesis/min in embryos treated with Y-27632.

Similar to our previous findings (Yu et al., 2021), myosin inhibition *in vivo* disrupted the ME boundary. We quantified a rapid increase in boundary roughness, with an initial 42±9% increase within 10 min ($P<0.01$), and a smaller secondary increase of 24±14% over the next 30 min, for a total increase of 66±14% by 40 min ($P<0.001$, Fig. 2A,B,E-G). Additionally, disrupting

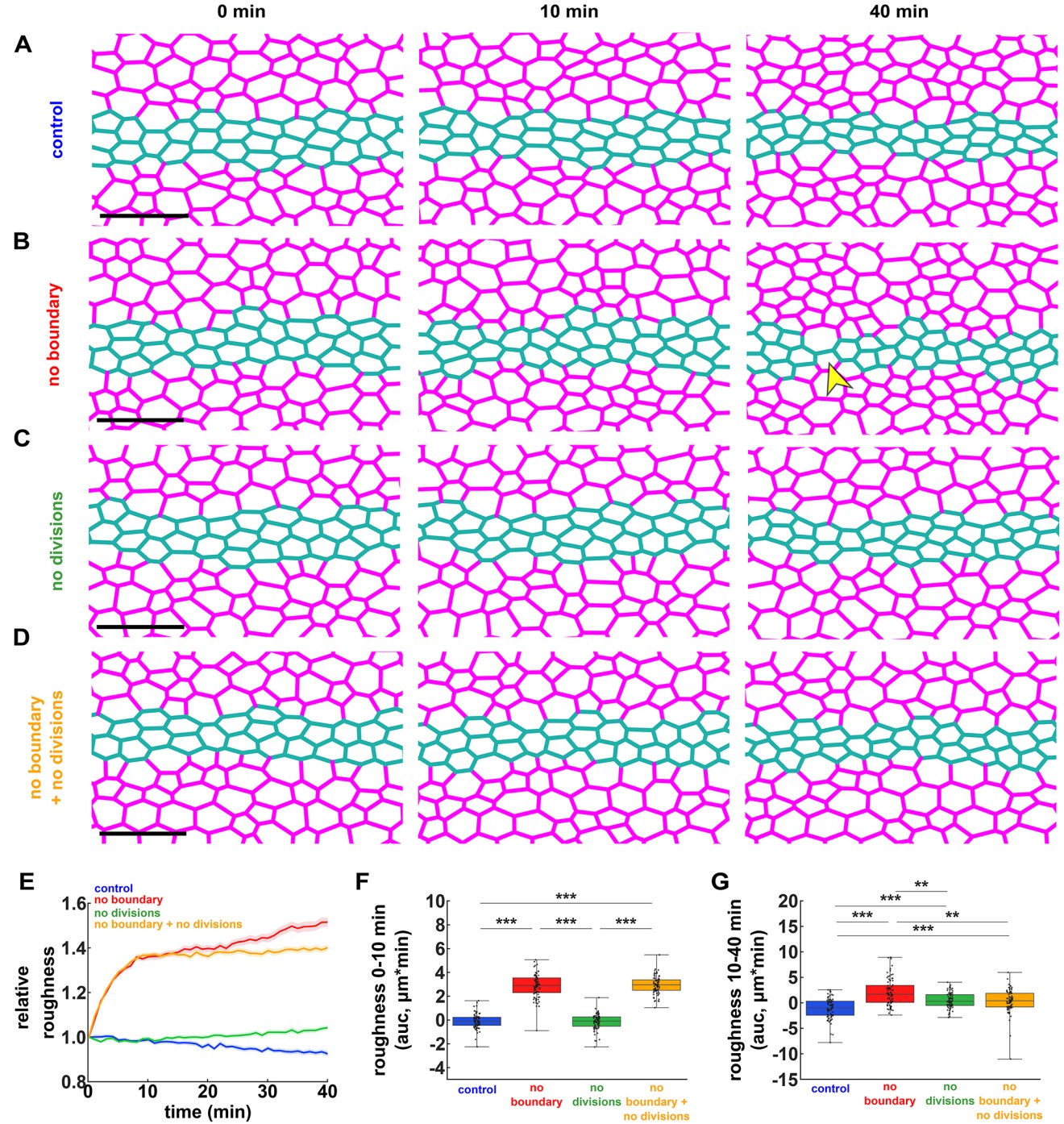

**Fig. 1. Mathematical modelling predicts that ectoderm divisions both challenge and refine the ME boundary.** (A-D) Simulations of mesectoderm ingression in control embryos (A), with an acute loss of tension (no boundary) at the ME interface (B), without ectoderm cell divisions (C), or with simultaneous acute loss of tension at the ME interface and inhibition of ectoderm cell divisions (D). Magenta indicates ectoderm; teal indicates mesectoderm. Arrowhead indicates contralateral ectoderm cells in close proximity. Scale bars: 20 µm. Anterior is towards the left. Time zero corresponds to the time at which the mesectoderm width starts decreasing. (E-G) Relative boundary roughness (E), and integrated changes in boundary roughness (area under the curve, auc) from 0-10 min (F) or from 10-40 min (G) in simulations of control embryos (blue, $n$=80 simulations), embryos with acute loss of tension at the ME boundary (red, $n$=80), embryos without ectoderm cell divisions (green, $n$=80), and embryos with both acute loss of tension at the ME boundary and no ectoderm cell divisions (orange, $n$=80). (E) Error bars indicate s.e.m. (F,G) Error bars indicate range; boxes indicate quartiles; grey lines indicate the median. **$P<0.01$, ***$P<0.001$ (Kruskal-Wallis and Dunn's tests).

the myosin cable at the ME boundary led to the formation of ectoderm bridges (Yu et al., 2021) (Fig. 2B, arrowheads). Thus, our results show that myosin activity is important to maintain a linear boundary and prevent cell mixing at the ME interface.

To determine if cell divisions deform the ME boundary when myosin is inhibited, we co-injected embryos with Y-27632 and dinaciclib after the ME boundary had formed, to simultaneously disrupt the ME boundary and prevent ectoderm cell divisions

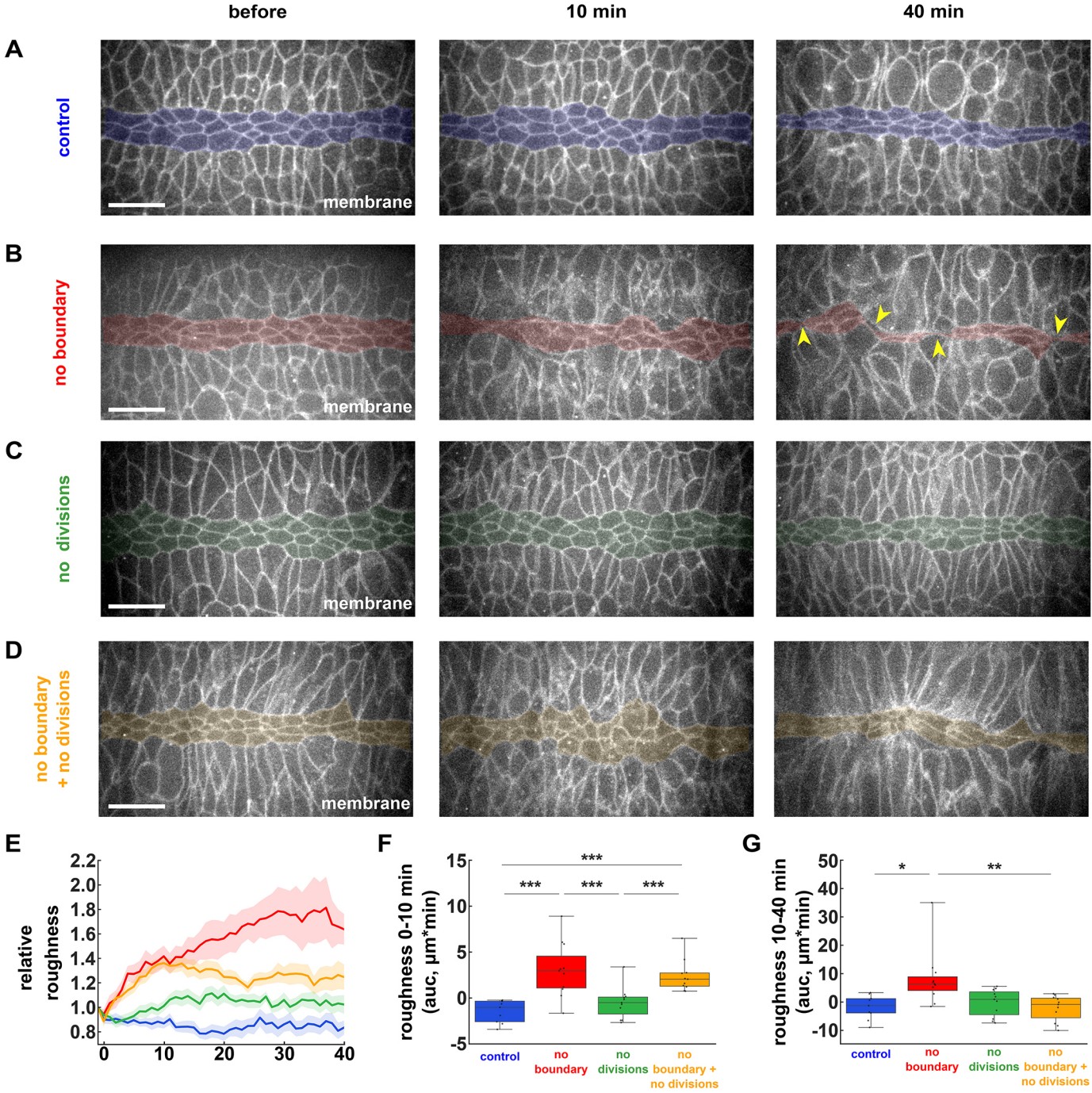

**Fig. 2. Ectoderm divisions both challenge and refine the ME boundary *in vivo*.** (A-D) Mesectoderm (centre, shaded) and ectoderm (top and bottom) cells in embryos expressing Gap43:mCherry and injected 1 h after the onset of mesectoderm divisions, with 50% DMSO (A), 20 mM Y-27632 (B), 500 µM dinaciclib (C), or both 20 mM Y-27632 and 500 µM dinaciclib (D). Arrowheads indicate contralateral ectoderm cells in close proximity. Scale bars: 20 µm. Anterior is towards the left. (E-G) Relative boundary roughness (E) and integrated changes in boundary roughness (area under the curve, auc) from 0-10 min (F) or from 10-40 min (G) in embryos treated with 50% DMSO (blue, *n*=9 embryos), 20 mM Y-27632 (red, *n*=11), 500 µM dinaciclib (green, *n*=11), or both 20 mM Y-27632 and 500 µM dinaciclib (yellow, *n*=11). (A-G) Time is given with respect to the time of injection. (E) Error bars indicate s.e.m. (F,G) Error bars indicate range; boxes indicate quartiles; grey lines indicate the median. *$P<0.05$, **$P<0.01$, ***$P<0.001$ (Kruskal-Wallis and Dunn's tests).

(Fig. 2A,D). Consistent with our *in silico* model predictions, inhibiting ectoderm divisions did not rescue the initial increase in boundary roughness associated with ME boundary disruption, but prevented the secondary increase in roughness at 40 min ($P<0.05$, Fig. 2E-G) and reduced the formation of ectoderm bridges. Together, our data indicate that ectoderm divisions *in vivo* challenge the ME

boundary, suggesting that the ME boundary resists ectoderm divisions to prevent cell mixing.

**Cell divisions contribute to ME boundary linearity**

Cell divisions typically challenge and deform tissue boundaries (Monier et al., 2010). Strikingly, our model predicted that cell

divisions may contribute to the maintenance of the ME boundary. In control simulations, the boundary was refined slowly but continuously, with roughness decreasing by 7±1% within 40 min (*P*<0.001, Fig. 1A,E), consistent with a better-defined interface between ectoderm and mesectoderm. Suppressing cell divisions in the model reverted this trend, with roughness increasing (rather than decreasing) by 4±1% (*P*<0.001, Fig. 1C,E-G). Thus, mathematical modelling suggests that cell divisions in the ectoderm may contribute to the linearity of the ME boundary.

To test whether cell divisions refine the ME boundary *in vivo*, we quantified the roughness of the ME boundary in control embryos and in embryos treated with dinaciclib to inhibit cell division. Consistent with model predictions, we found that, in control embryos, boundary roughness decreased significantly by 22±6% over 40 min (*P*<0.01, Fig. 3A,C,D and Movie 5). In contrast, inhibiting cell division caused a transient increase in ME boundary roughness, which remained significantly higher than in controls (*P*<0.05, Fig. 3A-D and Movie 5). Together, our results show that cell divisions not only challenge, but also refine the tissue boundary between ectoderm and mesectoderm in the *Drosophila* embryo.

### Cell divisions facilitate ectoderm cell movement

Cell divisions can fluidize tissues, facilitating the reorganization of cells (Lenne and Trivedi, 2022). To investigate if cell divisions refine the ME boundary by promoting the rearrangement of cells, we measured the mobility of ectoderm cells in the presence or absence of cell divisions (Fig. 4A,B). We measured cell mobility using a self-overlap function that characterizes glassy dynamics in molecular and colloidal materials (Castellani and Cavagna, 2005). Specifically, the self-overlap function measures the fraction of 'static cells' or cells that moved a distance shorter than a characteristic length scale over a time period (see Materials and Methods). In our computational model, the degree of self-overlap in the ectoderm in the presence of cell divisions decreased continuously, with half of the reduction (the half-time of self-overlap decrease) occurring by 31.1±0.4 min (*P*<0.001, Fig. 4A,C). In contrast, when we inhibited cell divisions in the model, the self-overlap function did not decrease for 83% of simulations (66/80),

and for the remaining simulations (14/80) the half-time of self-overlap reduction was 37.7±0.5 min (Fig. 4B,C), significantly longer than control simulations (*P*<0.001), suggesting that the cells were frozen when cell divisions were inhibited. We obtained similar results when we used the mean-squared displacement (MSD) of the cell centroids to quantify cell movements with respect to the mesectoderm: in controls, the MSD of ectoderm cells at 40 min decreased by 83% when cell divisions were inhibited (*P*<0.001, Fig. 4D-F). Thus, mathematical modelling predicts that cell divisions increase cellular mobility in the ectoderm.

To test the prediction that cell divisions increase cellular mobility *in vivo*, we calculated the degree of self-overlap for ectoderm cells in control embryos or in embryos in which cell divisions had been blocked with dinaciclib (Fig. 5A,B). Following dinaciclib injection, cells that were already dividing completed their divisions. Thus, we began our analysis 15 min after injection. Consistent with our model predictions, ectoderm cells at least one cell diameter away from the ME boundary changed their positions rapidly in the presence of divisions, with a half-time of self-overlap decrease of 27.1±2.2 min (*P*<0.001, Fig. 5A,C). In contrast, and as anticipated by the model, inhibiting cell divisions limited cell mobility in the ectoderm: the half-time of self-overlap decrease was 32.9±1.0 min, a significantly longer time than in controls (*P*<0.05, Fig. 5B,C). Similarly, the MSD of the cell centroids decreased by 51% when we inhibited cell divisions (*P*<0.01, Fig. 5D-F), further suggesting that, *in vivo*, cell divisions fluidize the ectoderm.

We investigated how cell divisions affected the mobility of cells immediately adjacent to the ME boundary. Both ectoderm cells adjacent to the boundary and mesectoderm cells displayed reduced mobility in controls with respect to ectoderm cells away from the boundary (Fig. 5A,B, Figs S4A,B and S5A,B): the MSDs of cell centroids decreased by 61% in ectoderm cells adjacent to the boundary (*P*<0.05, Fig. 5D-F and Fig. S4D-F) and by 76% in the mesectoderm (*P*<0.01, Fig. 5D-F and Fig. S5D-F). Inhibiting cell divisions had a more modest effect in cells adjacent to the ME boundary when compared with ectoderm cells further away from the boundary, with no significant changes in the half time of self-overlap decrease or the MSD when cell divisions were inhibited

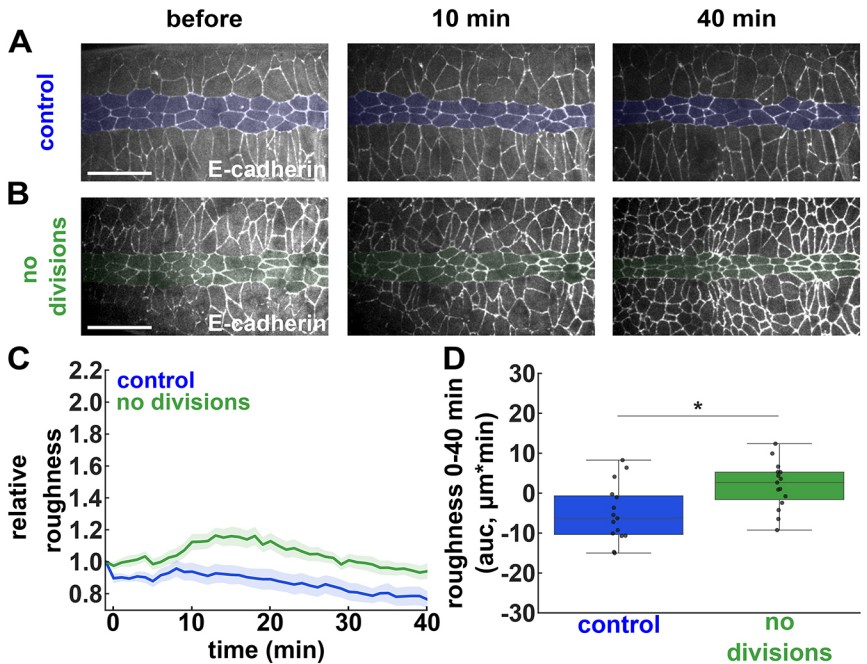

**Fig. 3. Ectoderm divisions increase the linearity of the ME boundary.** (A,B) Mesectoderm (centre, shaded) and ectoderm (top and bottom) cells expressing E-cadherin:GFP and injected 1 h after the onset of mesectoderm divisions with 50% DMSO (A) or 500 µM dinaciclib (B). Scale bars: 20 µm. Anterior is towards the left. (C,D) Relative boundary roughness (C) and integrated change in boundary roughness (area under the curve, auc) from 0-40 min after injection (D) in embryos treated with 50% DMSO (blue, *n*=9 embryos) or 500 µM dinaciclib (green, *n*=9). (A-D) Time is with respect to the time of injection. (C) Error bars indicate s.e.m. (D) Error bars indicate the range; boxes indicate quartiles; grey lines indicate the median. **P*<0.05 (Mann-Whitney test).

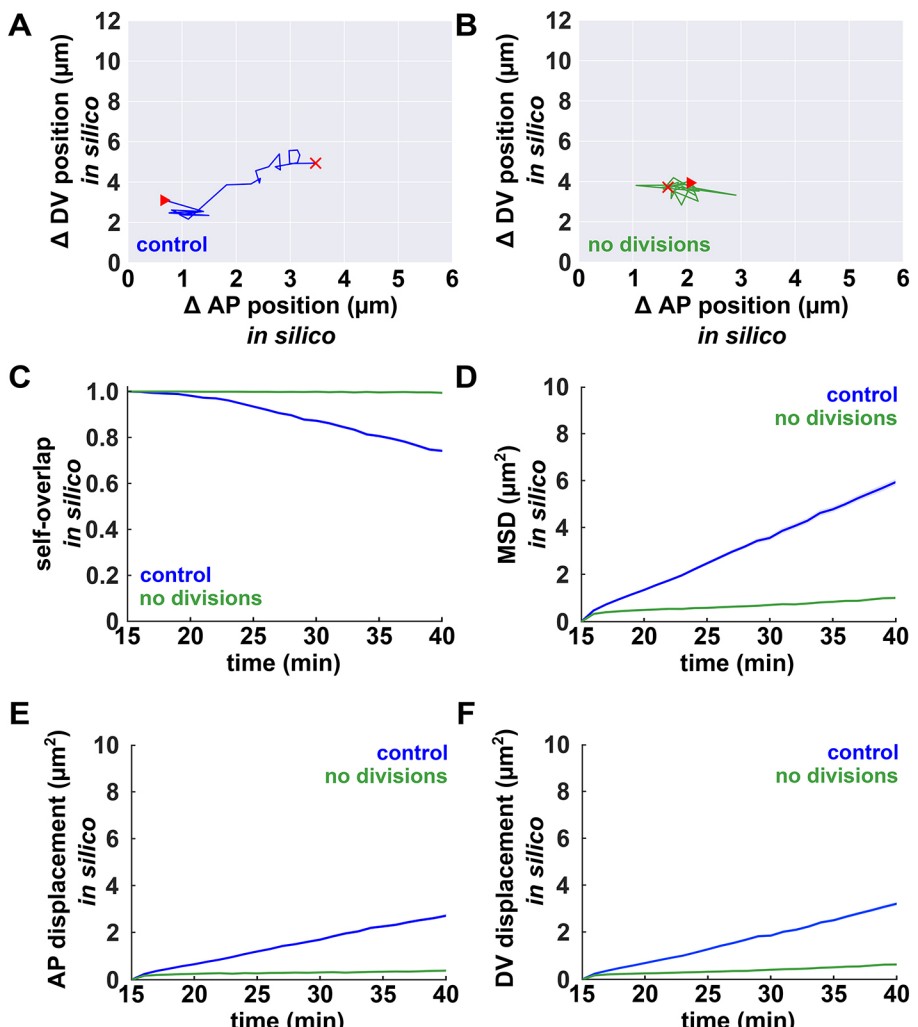

**Fig. 4. Mathematical modelling predicts that cell divisions increase ectoderm cell mobility.** (A,B) Sample ectoderm cell centroid trajectories *in silico*, for controls (blue) or when cell divisions were inhibited (green). Red triangles and crosses indicate the starting and final points of the trajectories, respectively. (C-F) Self-overlap function (C), MSD (D), mean squared anterior-posterior (AP) displacement (E) and mean squared dorsal-ventral (DV) displacement (F) for ectoderm cells *in silico*, in the presence (blue) or absence (green) of cell divisions (*n*=80 simulations per group, 40 cells per simulation). Error bars indicate s.e.m.

(Figs S4C and S5C). These results further suggest that ectoderm cell behaviours away from the ME boundary play an important role in promoting boundary refinement. Inhibiting cell divisions led to an increase in cell density *in vivo* (Fig. S6A-C), but not *in silico* (Fig. S6D-F), suggesting that changes in cell morphology are not responsible for the reduced cell mobility when divisions are inhibited. Of note, cell displacements when cell divisions were inhibited were greater *in vivo* that *in silico* (Figs 4D and 5D), possibly due to additional forces that contribute to the movement of ectoderm cells in living embryos, including intercalary cell behaviours in the ectoderm (Irvine and Wieschaus, 1994; Bertet et al., 2004; Blankenship et al., 2006), or the invagination of the mesoderm (Leptin and Grunewald, 1990; Sweeton et al., 1991; Clarke et al., 2025) and the posterior midgut (Collinet et al., 2015; Lye et al., 2015). Overall, our mathematical modelling and experimental results suggest that cell divisions may contribute to the refinement of the ME boundary by facilitating cellular movements.

## Cell divisions release tension and increase fluidity in the ectoderm

Cell divisions can release tissue tension (Streichan et al., 2014; Wang et al., 2017). We speculated that preventing cell divisions may increase tension in the ectoderm, which would in turn limit cell mobility. To test this possibility, we first used our mathematical model to compare ectoderm tension in the presence and absence

of cell divisions. We calculated junctional tension in the ectoderm directly from the energy of the cells (see Materials and Methods) (Movie 5). In the presence of cell divisions, tension at junctions between ectoderm cells decreased by 27.2±0.3% over the first 40 min of simulation (*P*<0.001, Fig. 6A,C,D). Inhibiting cell divisions had the opposite effect on junctional tension, which increased by 5.1±0.1% (*P*<0.01, Fig. 6C,D). Eventually junctional tension was 33% greater when cell divisions were inhibited than in controls (*P*<0.001, Fig. 6C). Together, our modelling results predict that cell divisions release tension in the ectoderm.

To test if cell divisions release ectoderm tension to facilitate cell remodelling *in vivo*, we used laser ablation to measure the tension sustained by individual cell-cell junctions in the ectoderm in both control and dinaciclib-treated embryos (Fig. 7A,B and Movie 6). Under the assumption of uniform viscoelastic properties, the initial recoil velocity after ablation of a cell-cell junction is proportional to the tension that the junction sustained (Hutson et al., 2003; Zulueta-Coarasa and Fernandez-Gonzalez, 2015). We found that the recoil velocity after ablation of ectoderm cell junctions away from the boundary was 24% greater when cell divisions were inhibited than in controls (*P*<0.05, Fig. 7A-C), in striking agreement with our model predictions. The increase in tension when cell divisions were inhibited was specific to junctions further away from the boundary (Fig. S7A-E), and was not associated with changes in the planar polarity or levels of myosin (Fig. S8A-E). Thus, our results indicate

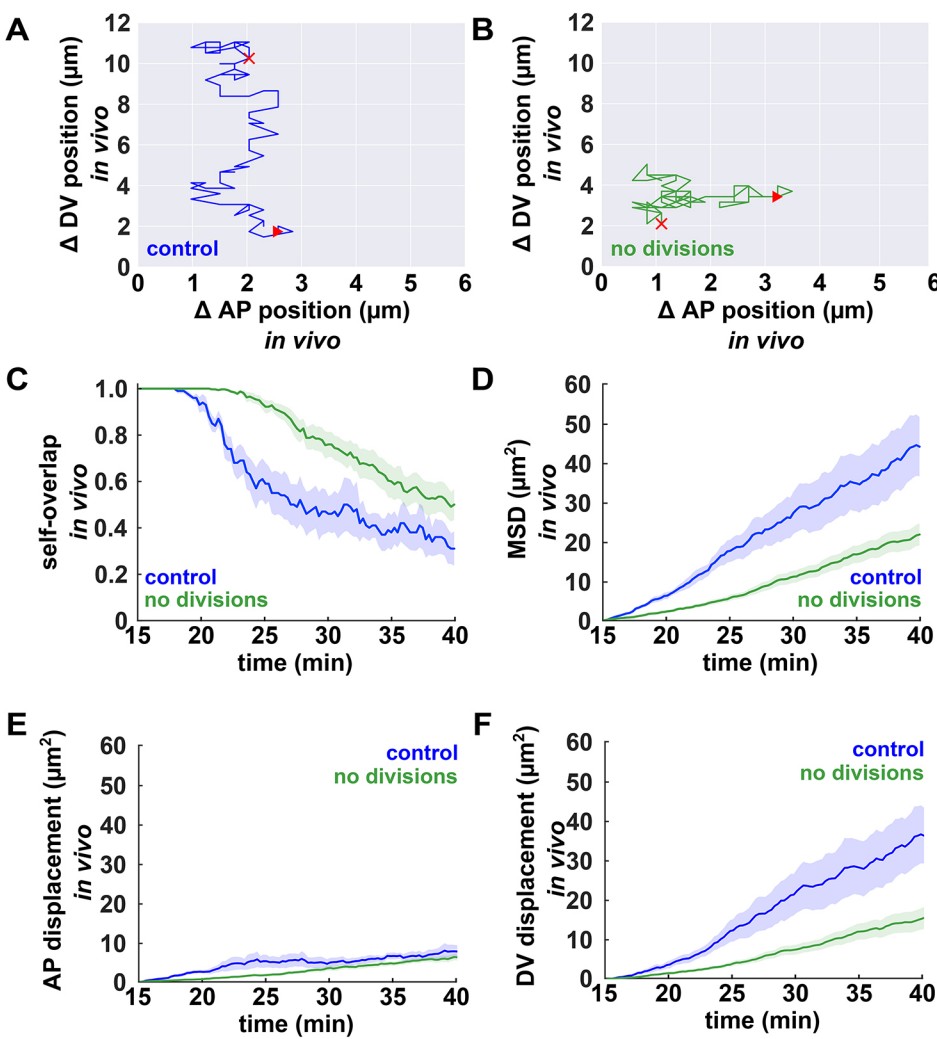

**Fig. 5. Cell divisions fluidize the ectoderm *in vivo*.** (A,B) Sample ectoderm cell centroid trajectories *in vivo* for controls (blue) or when cell divisions were inhibited (green). Red triangles and crosses indicate the starting and final points of the trajectories, respectively. (C-F) Self-overlap function (C), MSD (D), mean squared anterior-posterior (AP) displacement (E) and mean squared dorsal-ventral (DV) displacement (F) for ectoderm cells *in vivo* in DMSO-treated controls (blue, *n*=5 embryos, 20 cells per embryo) and in dinaciclib-treated embryos (green, *n*=16 embryos, 20 cells per embryo). Error bars indicate s.e.m.

that cell divisions reduce tissue tension in the ectoderm during mesectoderm internalization.

To further test the effect of cell divisions on the fluidity of the ectoderm, we used a Kelvin-Voigt mechanical equivalent circuit to fit the laser ablation results. The Kelvin-Voigt model allowed us to estimate a relaxation time that indicates how long it takes for the laser-induced displacements to dissipate, as well as the stress-to-elasticity ratio (see Materials and Methods). In agreement with our measurements of recoil velocity, the stress-to-elasticity ratio was 12% greater when cell divisions were inhibited (Fig. 7D), further suggesting that cell divisions reduce ectoderm tension. Additionally, we found that the relaxation time decreased by 21% when cell divisions were inhibited (*P*<0.001, Fig. 7E), consistent with 'freezing' of the tissue and the reduced cell mobility that we quantified. Together, our data indicate that cell divisions reduce junctional tension in the ectoderm, facilitating cell movement and tissue fluidity for ME boundary refinement.

## DISCUSSION

Boundaries must withstand challenges from forces generated during development to ensure proper tissue patterning and cell fate specification. Using mathematical modelling, we predicted that cell divisions within the ectoderm challenge the ME boundary. Surprisingly, our modelling also predicted that ectoderm divisions refine the ME boundary by releasing tension and fluidizing the

tissue. *In vivo* experiments using pharmacological treatments, quantitative microscopy and laser ablation support modelling predictions, showing that ectoderm divisions play dual roles both challenging and refining the ME boundary as the embryo develops. Our results also show that adaptive vertex models with tissue-specific features from *in vivo* measurements and time-varying parameters are highly effective in capturing complex active processes during embryonic development (Tah et al., 2025).

The mechanisms that control myosin localization at the ME boundary are unclear. Tension is partly responsible for the polarized localization of myosin to the interface between ectoderm and mesectoderm (Yu et al., 2021), although the origin of that tensile force is not known. Posterior pulling forces, generated by the invagination of the posterior midgut could generate anisotropic stress along the anterior-posterior axis, dissipated by ectoderm cells through intercalation (Collinet et al., 2015; Lye et al., 2015), but not by mesectoderm cells, which do not exchange neighbours (Wang et al., 2017). This model would thus predict that mesectoderm cells would be the main contributors to the accumulation of myosin at the ME boundary, something that has not been tested. Intercalary cell behaviours contribute to the convergent extension of the ectoderm (Irvine and Wieschaus, 1994; Bertet et al., 2004; Zallen and Zallen, 2004; Blankenship et al., 2006), which generates further anisotropic stress (Rauzi et al., 2008; Fernandez-Gonzalez et al., 2009) that may localize myosin at the ME boundary. Beyond tension-based myosin

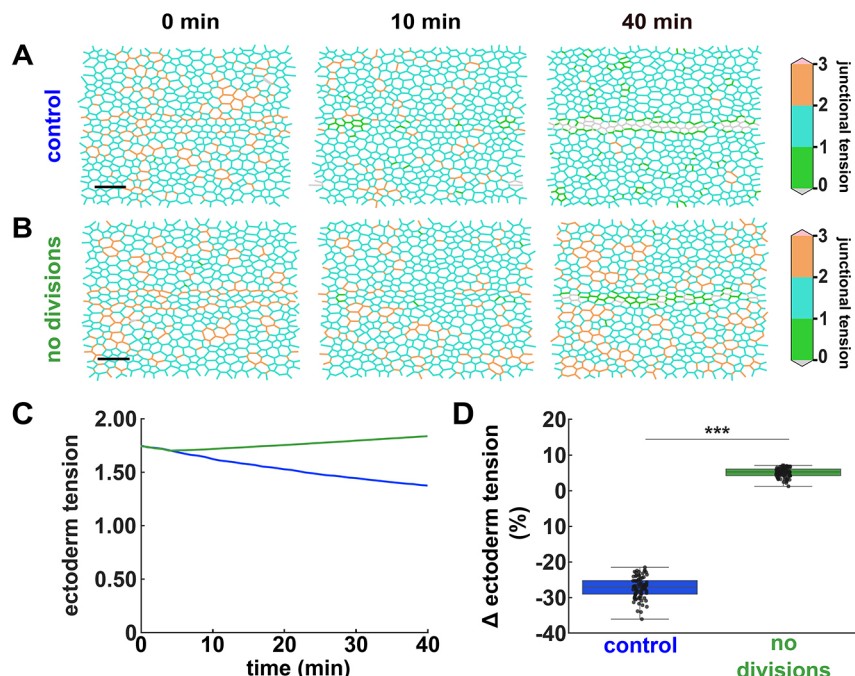

**Fig. 6. Mathematical modelling predicts that cell divisions reduce junctional tension.** (A,B) Junctional tension distribution in simulations of mesectoderm ingression in controls (A) or when ectoderm cell divisions were inhibited (B). Grey junctions are compressed. Scale bars: 20 μm. Anterior is towards the left. (C,D) Junctional tension over time (C) and percentage change in tension at 40 min (D) for ectoderm cells in control simulations (blue, $n$=80 simulations, 1439 junctions on average per simulation) or in simulations with no ectoderm divisions (green, $n$=80 simulations, 1234 junctions on average per simulation). (C) Error bars indicate s.e.m. (D) Error bars indicate the range; boxes indicate quartiles; grey lines indicate the median. ***$P$<0.001 (Mann-Whitney test).

localization, other mechanisms could drive the polarization of myosin at the ME boundary. The Eph/Ephrin receptor-ligand system promotes boundary formation. Eph/Ephrin signalling is associated with actomyosin accumulation at tissue borders (Cortina et al., 2007; Calzolari et al., 2014; Kindberg et al., 2021). In axons, Eph/Ephrin signalling activates the Rho GEF Ephexin via tyrosine phosphorylation, thus inducing Rho signalling and downstream actomyosin contractility (Sahin et al., 2005; Klein, 2012). Ephexin signalling also induces Rho signalling, actomyosin cable assembly and the formation of a boundary around damaged cells in developing embryos (Rothenberg et al., 2023). Importantly, overexpression of Ephrin in the mesectoderm causes defects in

the ventral nerve cord (Bossing and Brand, 2002), suggesting that Eph/Ephrin signalling may contribute to proper mesectoderm development.

Our data indicate that cell divisions not only challenge, but also refine the ME boundary. Increased tension at compartment boundaries often results in smooth interfaces with reduced roughness (Landsberg et al., 2009; Monier et al., 2010; Calzolari et al., 2014). However, the ME boundary becomes smoother while myosin levels decrease (Yu et al., 2021). Theoretical studies suggest that the frequency of cell divisions influences the relaxation time of a viscoelastic tissue: tissues undergoing more divisions behave in a more fluid manner (Ranft et al., 2010), as cell divisions disrupt the solid-like structure of the

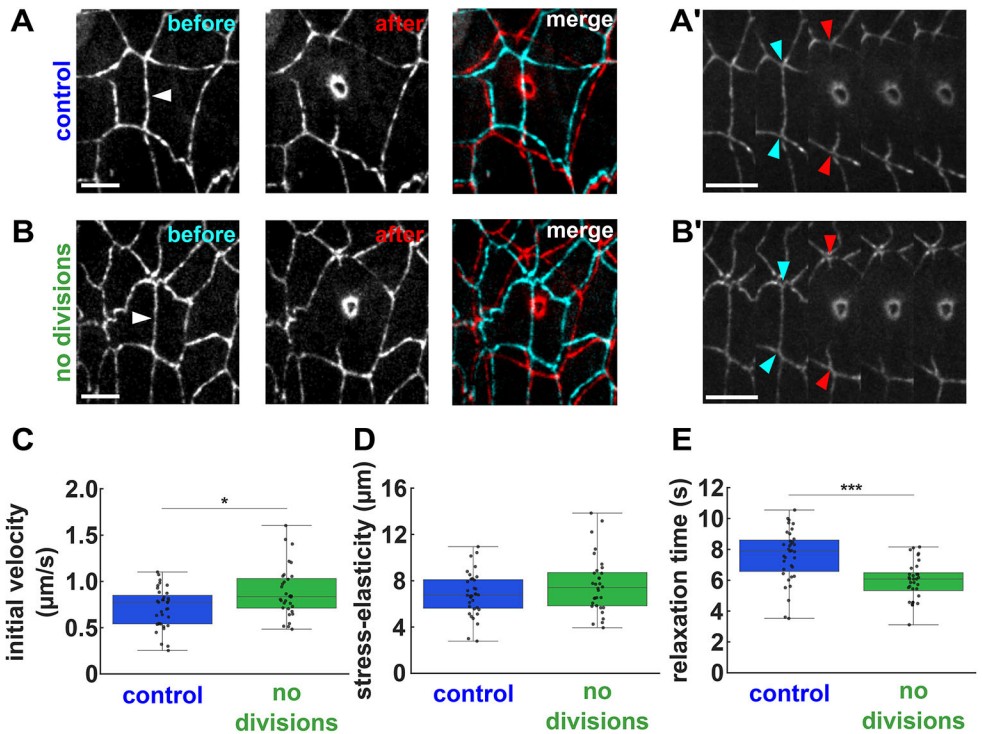

**Fig. 7. Cell divisions reduce junctional tension in the ectoderm *in vivo*.** (A,B) Ectoderm cells expressing E-cadherin:GFP immediately before (left, cyan in merge) and after (right, red in merge) ablation of a cell-cell junction parallel to the dorsal-ventral axis, in embryos treated with 50% DMSO (A) or 500 μM dinaciclib (B). (A′,B′) Corresponding kymographs are shown. Arrowheads indicate the severed interface (white, A,B), or its ends prior to ablation (cyan, A′,B′) or immediately after (red, A′,B′). Scale bars: 5 μm in A,B; 4 s in A′,B′. Anterior is towards the left. (C-E) Initial recoil velocity after ablation (C), stress-elasticity ratio (D) and relaxation time (E) for cuts in embryos treated with 50% DMSO (blue, $n$=35 cuts) or 500 μM dinaciclib (green, $n$=32). Error bars indicate the range; boxes indicate quartiles; grey lines indicate the median. *$P$<0.05, ***$P$<0.001 (Mann-Whitney test).

tissue. In gastrulating zebrafish embryos, oriented cell divisions within the plane of the enveloping layer alleviate tension and support tissue spreading (Campinho et al., 2013). Similarly, cell divisions in the mesectoderm release tension and facilitate axis elongation (Wang et al., 2017). We show that cell divisions reduce junctional tension in the ectoderm. Thus, by releasing tension and increasing ectoderm fluidity, cell divisions may enable cell movements that sharpen the ME boundary despite the loss of myosin. Our model also predicted that mesectoderm cell-cell contacts sustain compressive stress in controls. In contrast, when ectoderm divisions were inhibited, mesectoderm junctions predominantly sustained tensile stress, consistent with our data that ectoderm divisions push against the ME boundary, and suggesting that polarized myosin cables in the ectoderm may pull on the ME boundary, locally increasing boundary roughness and resisting mesectoderm internalization.

The mechanisms by which cell divisions reduce tissue tension in the ectoderm are unclear. Anisotropic strain can orient the mitotic spindle parallel to the axis of maximum deformation, in a process that releases strain (Campinho et al., 2013; Wyatt et al., 2015). However, ectoderm cells divide with random orientations, except those immediately adjacent to the ME boundary, which divide with their spindle perpendicular to the boundary (Yu et al., 2021). Thus, it is possible that mesectoderm internalization generates a local, dorsal-ventral-oriented pulling force that orients the divisions of ectoderm cells adjacent to the ectoderm, relaxing strain and tension throughout the tissue. Additionally, in tissues with low levels of proliferation, cells can adopt a solid-like or jammed state as cell junctions mature (Garcia et al., 2015), making rearrangements less feasible (Lawson-Keister and Manning, 2021). Thus, cell divisions in the ectoderm may destabilize cell-cell junctions, possibly increasing actomyosin turnover and limiting tension (Fernandez-Gonzalez et al., 2009). Studies quantifying how blocking cell divisions in the ectoderm affects the turnover of adherens junctions and cortical actomyosin may shed light on how cell division facilitates cell rearrangements for boundary refinement. It is important to note that tissue fluidity could increase through multiple mechanisms, including changes in cell-cell and cell-substrate interactions (friction) (Founounou et al., 2021; Staddon et al., 2022), Brownian noise (temperature) (Bi et al., 2016; Devany et al., 2021) or cell elongation (target cell shape) (Bi et al., 2015). Cell divisions perpendicular to the plane of the tissue could also fluidize tissues. Future work should extend and take advantage of vertex models to determine how different mechanisms that control tissue fluidity affect boundaries, to establish the uniqueness of the role of cell division on boundary maintenance.

In conclusion, our results suggest a dual role for ectoderm divisions on the ME boundary. Similar to other systems, divisions in the ectoderm challenge the ME boundary. Surprisingly, ectoderm divisions also refine the boundary. Thus, proliferating epithelia may maintain pre-established boundaries when contractile cables are disassembled, for instance at the dorsal-ventral boundary in the wing disc of late *Drosophila* larvae (Monier et al., 2011). The mechanisms that establish and maintain tissue boundaries are conserved (Sánchez-Corrales and Röper, 2018). Thus, our findings may reveal a conserved mechanism whereby boundaries across proliferative tissues not only resist, but also benefit from cell divisions.

### Limitations
Acute, tissue-specific inhibition of cell divisions in a living embryo is difficult. We used carefully timed pharmacological treatments that allowed mesectoderm cells to divide prior to inhibiting ectoderm cell divisions. But this approach could not specifically target cell

divisions adjacent to or far from the ME boundary. Optogenetic approaches that allow regional inhibition of cell division will enable validation of modelling predictions with regards to the relative contribution of different ectoderm populations to boundary refinement. Additionally, our results indicate that cell divisions modulate ectoderm tension without affecting myosin levels or distribution. Whether the effect of cell division on ectoderm tension is produced by controlling myosin turnover, or features of actin dynamics, remains to be determined.

## MATERIALS AND METHODS
### Fly stocks
We used the following markers for live imaging: *sqh-gap43:mCherry* (Martin et al., 2010), *sqh-sqh:GFP* (Royou et al., 2004) and *endo-e-cadherin:GFP* (Huang et al., 2009).

### Time-lapse imaging
Stage 7-9 embryos were dechorionated in 50% bleach for 2 min, rinsed, glued ventral side down to a glass coverslip using heptane glue, and mounted in a 1:1 mix of halocarbon oil 27 and 700 (Sigma-Aldrich). Embryos were imaged using a Revolution XD spinning disc confocal microscope (Andor Technology) equipped with an iXon Ultra 897 camera (Andor Technology). The dynamics of mesectoderm internalization as well as the effects of drug treatments were imaged with a 60× oil immersion lens (Olympus, NA 1.35). Sixteen-bit *z*-stacks were acquired at 0.5 µm steps every 4-60 s (15-27 slices per stack) and maximum intensity projections were used for analysis.

### Drug injections
Embryos were dechorionated and glued to a coverslip as above, dehydrated for 7.5 min, and covered with a 1:1 mix of halocarbon oil 27 and 700. Embryos were injected using an M-LSM motorized micromanipulator (Zaber) and a PV820 microinjector (WPI) attached to a spinning disc confocal microscope. Drugs (Y-27632, Tocris Bioscience; dinaciclib, ApexBio) were injected into the perivitelline space, where they are predicted to be diluted 50-fold (Foe and Alberts, 1983). Y-27632 was injected at 10 mM in 50% DMSO, and dinaciclib was injected at 500 µM in 50% DMSO; control embryos were injected with 50% DMSO. Drugs were injected 1 h after the first mesectoderm division.

### Laser ablation
Ablations were induced using a Micropoint pulsed N₂ laser (Andor) tuned to 365 nm. The laser delivers 120 µJ pulses of 2-6 ns each. Ten pulses were delivered at a single point to sever cell-cell junctions in the ectoderm. Samples were imaged immediately before and every 4 s after ablation. Laser cuts were conducted 30 min following drug treatment.

To estimate changes in viscoelasticity, we modelled cell-cell contacts as viscoelastic elements using a Kelvin-Voigt mechanical-equivalent circuit (Zulueta-Coarasa and Fernandez-Gonzalez, 2015). The Kelvin-Voigt circuit represents junctions as the combination of a spring (elasticity) and a dashpot (viscosity) configured in parallel. Considering the equations that represent the forces sustained by a spring and a dashpot, it is possible to derive the equation for the change in length between the ends of the junction after ablation:

$$L(t) = \frac{\sigma_0}{E}\left(1 - e^{-t\left(\frac{E}{\mu}\right)}\right) = D\left(1 - e^{-\frac{t}{\tau}}\right), \tag{1}$$

where $L(t)$ is the distance between the ends of the ablated junction at time $t$ after ablation, $\sigma_0$ is the tension sustained by the junction, $E$ is the elastic coefficient and $\mu$ is the viscosity. Using the laser ablation data it is possible to estimate the asymptotic value of $L$, $D$ and a relaxation time, $\tau$, that estimates the viscosity-to-elasticity ratio.

### Image segmentation and analysis
Image analysis was performed using our open-source image analysis platforms PyJAMAS (Fernandez-Gonzalez et al., 2022) and SIESTA (Fernandez-Gonzalez and Zallen, 2011; Leung and Fernandez-Gonzalez,

2015). To segment mesectoderm boundaries, we used the LiveWire algorithm in PyJAMAS, an interactive method based on Dijkstra's minimal cost path algorithm (Dijkstra, 1959), to find the brightest pixel path between any two pixels in an image. Cell boundaries were segmented using a combination of the LiveWire algorithm and the active contour method *balloons*, implemented in PyJAMAS, in which a polygon evolves on an image towards its minimum energy configuration, with the polygon energy inversely related to the image gradient and with a balloon force that ensures polygon expansion over regions of the image with small image gradients (Zulueta-Coarasa et al., 2014).

To measure the linearity of the ME boundary, we annotated the interface between ectoderm and mesectoderm using the LiveWire algorithm. The LiveWire annotation was rotated so that a fit line was horizontal, and detrended so that the mean Y-coordinate was zero. We quantified the roughness of the boundary as the standard deviation of the Y-coordinates of the pixels on the boundary after rotation and detrending. To assess the statistical significance of differences in roughness over time, we used the area under the curve as the test statistic. Area under the curve values were with respect to the value at the initial time point in the comparison (e.g. 10 min when comparing changes in roughness between 10 and 40 min after a drug treatment).

To measure cell movement in the ectoderm, we quantified a self-overlap function, $Q$ (Castellani and Cavagna, 2005), representing the fraction of cells at time $t$ that moved by less than 4 µm (approximately one cell radius; Yu et al., 2021):

$$Q(t) = \frac{1}{N} \sum_{i=1}^{N} w(|\mathbf{r}_i(t) - \mathbf{r}_i(0)|), \qquad (2)$$

where $\mathbf{r}_i$ is the position of $i$-th cell centroid, and $w$ is a step function that weighs the change in cell centroid position:

$$w(r) = \begin{cases} 1 & if \ r \leq 4\,\mu m \\ 0 & if \ r > 4\,\mu m \end{cases}. \qquad (3)$$

We validated our self-overlap results using the mean squared displacement of ectoderm cell centroids, defined as:

$$MSD(t) = \frac{1}{N} \sum_{i=1}^{N} |\mathbf{r}_i(t) - \mathbf{r}_i(0)|^2. \qquad (4)$$

The MSD was measured in time-lapse sequences registered to the position of the mesectoderm, to minimize cell movements associated with germband elongation and/or retraction, which are not included in our model (Irvine and Wieschaus, 1994; Schöck and Perrimon, 2002). Only ectoderm cells that had not divided were used for the self-overlap and MSD analyses. To quantify the mean squared anterior-posterior or dorsal-ventral displacements, we used the absolute value of the anterior-posterior or dorsal-ventral components, respectively, of the $\mathbf{r}_i(t) - \mathbf{r}_i(0)$ vector in Eqn 4.

We measured myosin polarity and levels in images of embryos expressing myosin:GFP. We rotated the images so that the anterior-posterior axis of the embryo was parallel to the horizontal axis of the image. We used OrientationJ (Rezakhaniha et al., 2012) to quantify myosin polarity. We measured the direction of the image gradient computed in a 3×3 pixel window, and assigned to each pixel an orientation perpendicular to that of the gradient. We summarized the degree of myosin polarization as the ratio of pixels oriented along the dorsal-ventral axis (75-90°) to pixels oriented along the anterior-posterior axis (0-15°). To quantify myosin fluorescence, we used a local threshold (Phansalkar et al., 2011) to isolate junctional pixels, and we quantified mean junctional fluorescence. The image median was used to subtract background.

## Vertex model
We used an adaptive 2D vertex model with cell divisions and time-varying parameters based on *in vivo* measurements. Our implementation utilized the open-source framework cell GPU (Sussman, 2017). In vertex models, cells are depicted as collections of nodes (vertices) and edges (interfaces between cells), which represent a cross-section of the tissue (Yu and Fernandez-Gonzalez, 2017). The energy, $E$, of the tissue is determined based on the geometry of the cells:

$$E = \sum_{i=1}^{N} [K_A(A_i - A_{0i})^2 + K_p(P_i - P_{0i})^2] + \sum_{\langle ij \rangle} \delta_{ij} \gamma_{ij} l_{ij}, \qquad (5)$$

where $N$ denotes the number of cells in the tissue. The first term of Eqn 5 is the incompressibility of the cell: $K_A$ is an area spring constant, and $A_i$ and $A_{0i}$ are the current and preferred areas of the $i$-th cell, respectively. The second term represents the competition between adhesion and contractility: $K_P$ is a perimeter spring constant, and $P_i$ and $P_{0i}$ are the current and preferred perimeters of the $i$-th cell, respectively. The final term represents the myosin cable between mesectoderm and ectoderm cells: $\delta_{ij}$ is 1 when the adjacent cells $i$ and $j$ belong to different tissue types, otherwise $\delta_{ij}$ is zero; $\gamma_{ij}$ represents the tension generated by the myosin cable at the interface between ectoderm and mesectoderm cells; and $l_{ij}$ denotes the length of the contact between cells $i$ and $j$.

The evolution of the model is guided by an energy minimization process. We used the forward Euler method to update the position of each cell vertex:

$$\Delta \mathbf{r}_k = \mu \mathbf{F}_k \Delta t + \boldsymbol{\eta}_k, \qquad (6)$$

where $\mathbf{r}_k$ is the position of vertex $k$, $\mathbf{F}_k = -\nabla_k E$ is the force on vertex $k$, $\mu$ is the inverse friction coefficient, $\Delta t$ is the integration time step, and $\boldsymbol{\eta}_k$ is a normally distributed random force with zero mean and variance $2\mu T \Delta t$. The temperature $T$ represents the strength of the Brownian noise in the position of cell vertices (Braňka and Heyes, 1999). We used $T$=0.010. The natural unit length of the simulations is given by $l = \sqrt{A_0}$. We set the integration time step $\Delta t$=0.01$\tau$, where $\tau$=1/$(\mu K_A A_0)$ is the natural time unit of the simulations.

We initialized the simulations with 400 random cells in a periodic box. The initial configurations were generated using Voronoi tessellations of randomly distributed points. All simulations were run at their target parameters for $10^3 \tau$ before setting ectoderm and mesectoderm cells features. We simulated the system after the completion of cell divisions in the mesectoderm and the formation of the myosin cables at the ME boundary. Thus, we set a common preferred area, $A_{0mesec}$, for all mesectoderm cells (Yu et al., 2021) (Table S1). The preferred cell area for mesectoderm cells, $A_{0mesec}$, was updated over time based on experimental data to simulate apical constriction $A_{0mesec} = A_{0mesec}^{initial} - 0.005 \times t$ (Yu et al., 2021). We modelled the ectoderm as a bidisperse mixture of cells representing before and after division, with preferred areas, $A_0 ecto^{pre-division}$ and $A_0 ecto^{post-division}$, respectively (Yu et al., 2021) (Table S1). We scaled the preferred perimeter, $P_0$, for all cell types to maintain a constant target shape index defined as $q = P_0/\sqrt{A_0}$ (Bi et al., 2015). We used a constant value of $q$=3.4 as the target shape index to scale the preferred perimeter for all cell types while allowing the cell type-specific cell size heterogeneity observed *in vivo*. We modelled the tension reduction at the ME boundary cell-cell interfaces using an exponential decay function $\gamma_{ij} = \gamma_0 e^{-k_\gamma t}$, where $\gamma_0$ is a constant line tension representing the myosin cable at the boundary at $t$=0, and $k_\gamma$ is the rate of tension reduction. We select $k_\gamma$ values based on experimental myosin measurements at the boundary in control embryos and in embryos in which myosin activity was inhibited (Yu et al., 2021) (Table S1). Ectoderm cells were randomly chosen to divide, with frequency and cell division orientation determined experimentally (Yu et al., 2021). Following each division, the system was allowed to relax to a new energy minimum. To model the impact of inhibiting the cell divisions, ectoderm cell divisions were blocked at 5 min in the simulations to mimic the delayed effect of dinaciclib injection. Other parameter values (Table S1) were selected to minimize the difference between *in vivo* and *in silico* measurements of ME boundary roughness. We assigned the natural time unit of the simulation to be $\tau$=0.1 min.

We calculated junctional tension in the model directly from the energy of the cells (Yang et al., 2017):

$$T_{ab} = \frac{\partial E}{\partial l_{ab}}, \qquad (7)$$

where $T_{ab}$ is the tension along the edge that connects vertices $a$ and $b$. Using Eqn 7, junctional tension in the ectoderm becomes:

$$T_{ab}^{ect} = 2K_P((P_j - P_{0j}) + (P_k - P_{0k})), \qquad (8)$$

where $a$ and $b$ are the vertices that connect cells $j$ and $k$.

## Statistical analysis

For multiple group comparisons, we used a Kruskal–Wallis test to reject the null hypothesis, followed by Dunn's test for pairwise comparisons (Glantz, 2002). Dividing and non-dividing cells were compared using a non-parametric Mann–Whitney test, or Wilcoxon signed-rank test for paired data. For time series, error bars indicate the s.e.m. For box plots, error bars show the range, the box indicates the quartiles and grey lines denote the median.

## Acknowledgements

We are grateful to Ashley Bruce and Ulli Tepass for useful discussions. FlyBase provided important information for this study.

## Competing interests

The authors declare no competing or financial interests.

## Author contributions

Conceptualization: V.C.; Data curation: V.C., R.F.-G., G.E.-T.; Formal analysis: V.C., M.M., R.P.-V., R.F.-G., G.E.-T.; Funding acquisition: R.F.-G., G.E.-T.; Investigation: V.C., M.M., R.F.-G., G.E.-T.; Methodology: R.P.-V., R.F.-G., G.E.-T.; Project administration: R.F.-G., G.E.-T.; Resources: R.F.-G., G.E.-T.; Software: R.F.-G., G.E.-T.; Supervision: R.F.-G., G.E.-T.; Validation: V.C., R.P.-V.; Visualization: V.C., M.M., G.E.-T.; Writing – original draft: V.C., R.F.-G., G.E.-T.; Writing – review & editing: V.C., M.M., R.P.-V., R.F.-G., G.E.-T.

## Funding

Our research was supported by grants to R.F.-G. from the Natural Sciences and Engineering Research Council of Canada (RGPIN-2025-04527), the Canada Foundation for Innovation (30279 and 43988), the Translational Biology and Engineering Program of the Ted Rogers Centre for Heart Research, and the Canadian Institutes of Health Research (156279 and 186188); and to G.E.-T. from the Natural Sciences and Engineering Research Council of Canada (DGECR-2023-00232) and the Western Strategic Support for Tri-Council Success-Seed Grant of Western University. R.F.-G. is the Canada Research Chair in Quantitative Cell Biology and Morphogenesis. Open Access funding provided by University of Toronto. Deposited in PMC for immediate release.

## Data and resource availability

All relevant data and details of resources can be found within the article and its supplementary information.

## The people behind the papers

This article has an associated 'The people behind the papers' interview with some of the authors.

## Peer review history

The peer review history is available online at https://journals.biologists.com/dev/lookup/doi/10.1242/dev.204817.reviewer-comments.pdf

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
