## [Peer Review File · Development (Cambridge, England)]

Cell divisions both challenge and refine tissue boundaries in the *Drosophila* embryo

Veronica Castle, Merdeka Miles, Rafael Perez-Vicente, Rodrigo Fernandez-Gonzalez and Gonca Erdemci-Tandogan
DOI: 10.1242/dev.204817

Editor: Paul Francois

Review timeline

Original submission:	27 March 2025
Editorial decision:	26 May 2025
First revision received:	16 December 2025
Accepted:	13 January 2026

Original submission

First decision letter

MS ID#: dev.204817

MS TITLE: Cell divisions both challenge and refine tissue boundaries in the *Drosophila* embryo

AUTHORS: Veronica Castle, Merdeka Miles, Rafael Perez-Vicente, Rodrigo Fernandez-Gonzalez and Gonca Erdemci-Tandogan

Dear Dr Fernandez-Gonzalez,

I have now received all the referees' reports on the above manuscript, and have reached a decision. The referees' comments are appended below, or you can access them online: please go to:

As you will see, the referees express considerable interest in your work, but have some significant criticisms and recommend a substantial revision of your manuscript before we can consider publication. If you are able to revise the manuscript along the lines suggested, which may involve further experiments, I will be happy receive a revised version of the manuscript. Your revised paper will be re-reviewed by one or more of the original referees, and acceptance of your manuscript will depend on your addressing satisfactorily the reviewers' major concerns. Please also note that *Development* will normally permit only one round of major revision. If it would be helpful, you are welcome to contact us to discuss your revision in greater detail. Please send us a point-by-point response indicating your plans for addressing the referees' comments, and we will look over this and provide further guidance.

Please attend to all of the reviewers' comments and ensure that you clearly highlight all changes made in the revised manuscript. Please avoid using 'Tracked changes' in Word files as these are lost in PDF conversion. I should be grateful if you would also provide a point-by-point response detailing how you have dealt with the points raised by the reviewers in the 'Response to Reviewers' box. If you do not agree with any of their criticisms or suggestions please explain clearly why this is so.

Reviewer 1

Advance summary and potential significance to field

This paper uses the mesectoderm/ectoderm (ME) boundary in *Drosophila* embryos to investigate the role cell division plays in challenging and developing tissue boundaries. Computational modelling and drug treatments (in vivo system) are used to manipulate both the contractile machinery known to help generate the boundary and divisions in the ectoderm. They show that, as previously known at other tissue boundaries, that cell divisions challenge the ME boundary. More surprisingly they conclude that divisions also help to define/develop the ME boundary by increasing the fluidity of the tissue (by releasing tissue tension). They argue this allows greater cellular movement, which in turn reduces the roughness of the boundary. This beneficial role of cell division in defining tissue boundaries is an interesting development in the field. However, there are some inconsistencies in the data that need to be addressed and beyond this the mechanistic link is missing between how the release in tension (and increase in fluidity) provided by cell division specifically reduces the roughness of the boundary.

Comments for the author

Major comments

- 1) Work in figures 1 and 2 demonstrate the divisions challenge the ME boundary but no attempt is made to understand the mechanism behind this - is it the rounding (/volume increase) process as shown in previous work (indeed is this aspect of division included in the model?)? Orientation of cell divisions? Ingression of the furrow at the boundary site? Do the divisions need to be specifically at the boundary site to have an effect or is it all divisions in the Ectoderm that have are damaging? Can the modelling be used to answer these questions?
- 2) In Figure 5 it is stated that the first 15 mins after injection are removed from the data as this is when any divisions in progress have completed. Is this done for all the movies using dinaciclib, if so why not? This is potentially important as in figure 3c (and Figure 2e) there is an initial increase in the "roughness" of the ME boundary during the first 15 mins for the "no division conditions" but afterwards both conditions appear very similar. Suggesting this first 15 mins after treatment might be important. Importantly this increase (and then decrease) was not seen in the modelling and suggests something is occurring that isn't accounted for in the model. The increase could be caused by an initial reaction to the drug treatment or the affect of cells preparing to divide and stalling. Is there a change in roughness between the two conditions if analysis starts once all cell division has ceased?
- 3) Does the cell mobility assessment distinguish between actual cell movement and cell shape change? Cell elongation in one direction for example would move the centroid of the cell without the cell actually moving. Indeed at 40 mins in figures 3 A and B the control cells appear to be larger and elongated perpendicularly to the ME boundary compared to the "no division" image. This analysis (in silico and in vivo) should be extended to an assessment of cell shape change and the direction of that change (AP vs DV).
- 4) Is the mobility/cell shape analysis limited to only the ectoderm cells? This should be extended to the mesectoderm cells too. Cells on both sides of the boundary will potentially have an effect on how "rough" it is.
- 5) In figures 5 A and B there is a clear difference in the cell movements (or potentially shape change) seen. The control cell moves mainly in the DV axis, while in the absence of cell divisions this movement appears to be mainly in the AP axis. An assessment of AP vs DV movement should be made as a directional movement/shape change would potentially have an effect on the forces acting on the ME boundary.
- 6) The orientation of cell divisions should be assessed - As they are linked to cell shape they will both be affected by and play a role in determining the direction the tissue grows in. This analysis should be separated into cells in contact with and not contacting the ME boundary to assess for local affects.
- 7) The analysis of tension distribution should also be carried out in the in vivo movies. Does it match the findings of the in silico analysis? This should be extended to direction of tension relative to AP and DV axis, not just general increase vs decrease. It should also be examined on a more local

level - how does tension in the cells immediately either side of ME boundary change compared to the tissue further away?

8) In the discussion it is suggested that divisions prevent the build up of junctional acto-myosin. This should be tested by staining for phosphorylated myosin.

Minor comments

- 1) Figure 1 A-D - are these "zoom-ins" of a larger simulation or do the configuration of the cells T0 vary? The starting images don't match. Can this be made clear please.
- 2) About figure 4 - "In our computational model, the degree of self-overlap in the ectoderm in the presence of cell divisions decreased continuously, reaching half of its final value by 31 min - I don't understand this? At 31min it's at about 0.85 and the final value is about 0.75 - which isn't half. Likewise for the statement about the treated sample too. This line doesn't drop at all - there is no half-time. Please can this be clarified.
- 3) Figure 6 A at 40 mins - there are grey cells in the mesectoderm - does this mean they under compression? Please clarify
- 4) In figure 6 there are also differences in the mesectoderm, these should also be discussed
- 5) Figure 7. Where were the ablation experiments done - were they on boundary cells or not? Were the junctions cut always AP or DV oriented etc... Please specify.
- 6) Figure 5 A and B - can a colour coding for time be added to the tracks please.

Reviewer 2

Advance summary and potential significance to field

The authors study the mesectoderm-ectoderm boundary of the early *Drosophila* embryo as a model system for tissue boundary maintenance. In line with previous findings, they show that acto-myosin generated tension stabilizes the boundary against fluctuations generated by cell divisions. Remarkably they find that in the presence of acto-myosin generated tension, cell divisions can refine (or sharpen) the boundary.

Overall, the manuscript is well written and will be of interest to the readership of *Development*. However, before I can recommend the manuscript for publication, the authors should address the following points.

The main claim is that cell divisions help increase tissue fluidity and thereby refine the tissue boundary. In the vertex model, there are several ways to increase tissue fluidity. In addition to increasing the cell division rate, the target shape index or the "temperature" (strength of force fluctuations), could be increased to drive a solid-to-fluid transition. These two variants should be tested in the model to demonstrate that it is indeed the increase in fluidity that helps sharpen the tissue boundary.

In the first paragraph of the results section the authors write "Ectoderm cells were randomly selected to divide, with a frequency and orientation based on experimental data (Yu et al., 2021)." It would be good if the authors would briefly mention what this frequency and orientation are to make the manuscript more self-contained. Moreover, the authors could test other division orientations in the model. Would divisions perpendicular to the boundary challenge but not refine it?

Experimentally, the authors could test whether changing the division axis, i.e. to become vertical (cleavage plane perpendicular to the apico-basal axis), changes the effect of cell divisions on the tissue boundary. One might hypothesize that vertical divisions would not challenge the boundary but might still help refine it because they might reduce tension in the ectoderm.

In the discussion, the authors state "the origin of that tensile force is not known." referring to the force acting along the mesectoderm-ectoderm (ME) tissue boundary. I don't believe this statement is fair in the light of the extensive literature on the tissue mechanics of germ band extension.

Active cell intercalations in the ectoderm drive extension of the germ band and thereby stretch the ME boundary. The authors should mention this key contribution to stretching of the ME boundary. In a similar vein, I don't think that the statement "The mechanisms by which cell divisions reduce tissue tension are unclear." is correct. The role of cell divisions on tissue mechanics has been

studied extensively. The main effect is that divisions allow the tissue to expand and therefore relax extensible tissue strain. Assuming that the junctions have some (effective) elastic properties, like the area and perimeter elasticity of the vertex model, relaxation of strain implies relaxation of stress. This should be mentioned by the authors.

Minor points:

The authors should justify their choice of model parameters, in particular the target shape index which plays a key role for the material properties of the vertex model tissue. The "temperature" should be called noise strength to avoid confusion with the actual temperature.

The measure of boundary roughness Eq. (2) could be justified better and made more systematic. Why is it measured along 13 μm segments? Are the results robust when changing the segment length? A more systematic way to measure the roughness would be tortuosity, commonly defined as the arclength divided by the end-to-end distance of a curve.

Typos:

In Eq. (8) it should read $P_i - P_{0i}$, instead of $P_i + P_{0i}$

Reviewer 3

Advance summary and potential significance to field

The paper establishes the mesoderm/ectoderm compartment boundary in the early fly embryo as a new experimental system for the quantitative analysis of the mechanisms by which developing organisms control borders between different cell and tissue types. The main observation is that cell divisions can both disrupt the straightness of the border and relieve tissue tension. These results are based on quantitative analysis of imaging data, laser ablation experiments, and pharmacological perturbations. The results are also supported by and, to a significant extent, supported by a clear model, based on the vertex description of the epithelium with an appropriate modification for region-specific control of tissue tensions. The manuscript was a pleasure to read. The authors did an excellent job motivating their work and presented results in a very logical way. I see no flaws in the study and am happy to recommend it for publication.

Comments for the author

A couple of suggestions: it would be good to discuss the robustness of the model fits and try to come up with a dimensionless group that can be used to classify different types of effects of cell division on boundary straightness. I would also like to see a more clear discussion of the differences of the general effects in this study and the work on compartment boundaries in the wing imaging disc, especially since that system was used to establish a lot of ideas and tools used by the authors. Finally, it would be great to have a more resolved view of local tissue geometry changes accompanying a mitotic event.

First revision

Author response to reviewers' comments

Reviewer 1: SUMMARY OF THE ADVANCE MADE IN THIS PAPER AND ITS POTENTIAL SIGNIFICANCE TO THE FIELD

This paper uses the mesoderm/ectoderm (ME) boundary in *Drosophila* embryos to investigate the role cell division plays in challenging and developing tissue boundaries. Computational modelling and drug treatments (in vivo system) are used to manipulate both the contractile machinery known to help generate the boundary and divisions in the ectoderm. They show that, as previously known at other tissue boundaries, that cell divisions challenge the ME boundary. More

surprisingly they conclude that divisions also help to define/develop the ME boundary by increasing the fluidity of the tissue (by releasing tissue tension). They argue this allows greater cellular movement, which in turn reduces the roughness of the boundary. This beneficial role of cell division in defining tissue boundaries is an interesting development in the field. However, there are some inconsistencies in the data that need to be addressed and beyond this the mechanistic link is missing between how the release in tension (and increase in fluidity) provided by cell division specifically reduces the roughness of the boundary.

SUGGESTIONS TO AUTHORS

Major comments

1) Work in figures 1 and 2 demonstrate the divisions challenge the ME boundary but no attempt is made to understand the mechanism behind this - is it the rounding (/volume increase) process as shown in previous work (indeed is this aspect of division included in the model?); Orientation of cell divisions? Ingression of the furrow at the boundary site? Do the divisions need to be specifically at the boundary site to have an effect or is it all divisions in the Ectoderm that have an effect? Can the modelling be used to answer these questions?

We used our mathematical model to investigate how cell divisions challenge the mesectoderm-ectoderm (ME) boundary. We note that, both *in silico* and *in vivo*, the initial increase in roughness (within 10 minutes) when tension is disrupted still happens in the absence of cell divisions (Figs. 1A, D, F and 2A, D, F), indicating that the initial increase in roughness is a consequence of the loss of the cable, and not of cell division. We previously showed that ectoderm cells move across the ME boundary when tension is inhibited (Yu et al., 2021), suggesting that cell movement, rather than cell division, is responsible for the initial increase in roughness when the boundary is disrupted. In contrast, the secondary increase in roughness when tension was abolished required cell division (Figs. 1A, D, G and 2A, D, G), and therefore we focused on that secondary increase.

Our modelling conditions can rule out potential mechanisms by which cell divisions challenge the boundary. In our model, cells do not increase their area or round up as they divide (Yu et al., 2021), suggesting that changes in the morphology of dividing cells are not responsible for disrupting the ME boundary.

To establish if apical constriction in the mesectoderm contributes to the secondary increase in ME boundary roughness when tension at the boundary decreased, we simulated scenarios with both a decrease in tension at the boundary and no divisions, as well as (a) a monotonically decreasing preferred area for mesectoderm cells, A_{0mesec} , defined as $A_{0mesec} = A_{0mesec}^{initial} - 0.005 * t$ (Yu et al., 2021) to simulate apical constriction, or (b) a constant value for A_{0mesec} that limited mesectoderm internalization (new Fig. S3). We found that limiting mesectoderm internalization did not alter the impact of cell division on the secondary increase in roughness, suggesting that the reduction in apical area of mesectoderm cells does not contribute to the increased roughness of the ME interface when actomyosin contractility is disrupted. We included the new data in the Results (page 6, paragraph 2, line 11):

"Importantly, limiting the apical constriction of mesectoderm cells in the simulations did not affect the increase in ME boundary roughness when actomyosin contractility was inhibited (Fig. S3A-B, E), or the rescue of the roughness phenotype when cell divisions were blocked (Fig. S3C-E), suggesting that changes in mesectoderm cell morphology do not challenge the ME boundary."

To determine if the orientation of ectoderm divisions challenges the ME boundary, we used our model to alter cell division orientation. *In vivo*, ectoderm cells adjacent to the ME boundary divide primarily with their spindle oriented perpendicular to the ME boundary, while ectoderm cells away from the boundary do not display a clear preference in spindle orientation (Yu et al., 2021). We used the *in vivo* distribution of division orientations in the current study. We have now clearly indicated how cell divisions were oriented in the Results (page 5, paragraph 2, line 10):

"Ectoderm cells were stochastically selected to divide, with a frequency and orientation based on experimental data (Yu et al., 2021): cells adjacent to the ME boundary oriented their spindle

preferentially along the dorsal-ventral axis, while the rest of the cells divided with random spindle orientations."

To investigate if ectoderm cell division orientation challenges the ME boundary, we randomized the orientation of the divisions of ectoderm cells adjacent to the boundary, and we quantified changes in roughness of the ME boundary both in the presence and absence of tension at the boundary (new Fig. S1). We found that altering the orientation of cell divisions in ectoderm cells adjacent to the boundary did not affect our simulation results, both in the presence and absence of tension. Together these results suggest that cell division orientation adjacent to the boundary is not a major challenge to the ME interface. We have included a description of these new data in the Results section (page 6, paragraph 2, line 1):

"To further establish how ectoderm cell divisions may impact the ME boundary, we simulated scenarios altering the orientation of cell divisions adjacent to the boundary. We found that randomizing the angle of division of ectoderm cells adjacent to the boundary (Fig. S1A-B, D-E, G), or rotating the angle by 90 degrees, such that the spindle was oriented preferentially along the anterior-posterior axis (Fig. S1A, C-D, F-G) did not affect the increase in roughness when tension at the boundary decreased, or the rescue of roughness when cell divisions were inhibited, consistent with the relatively small number of ectoderm cells adjacent to the boundary."

Lastly, we investigate whether cell divisions adjacent to the ME boundary or away from the boundary contribute differentially to the increase in roughness when tension is disrupted. To this end, we quantified boundary roughness in simulations in which tension at the ME interface had been suppressed and all divisions, divisions adjacent to the boundary, or divisions away from the boundary had been inhibited (new Fig. S2). We found that inhibiting cell divisions far from the boundary or decreasing their numbers matched the effects of inhibiting all divisions, suggesting that it is the number of dividing cells, rather than their position or orientation, that challenges the ME boundary. These new data have been included in the Results (page 6, paragraph 2, line 7):

"Instead, the rescue in boundary roughness in the absence of tension could only be accomplished when all cell divisions (Fig. S2A-B, F) or cell divisions far from the boundary (Fig. S2C-F) were inhibited or reduced in number, suggesting that the number of ectoderm cell divisions, rather than their position or orientation, challenges the ME interface."

2) In Figure 5 it is stated that the first 15 mins after injection are removed from the data as this is when any divisions in progress have completed. Is this done for all the movies using dinaciclib, if so why not? This is potentially important as in figure 3c (and Figure 2e) there is an initial increase in the "roughness" of the ME boundary during the first 15 mins for the "no division conditions" but afterwards both conditions appear very similar. Suggesting this first 15 mins after treatment might be important. Importantly this increase (and then decrease) was not seen in the modelling and suggests something is occurring that isn't accounted for in the model. The increase could be caused by an initial reaction to the drug treatment or the affect of cells preparing to divide and stalling. Is there a change in roughness between the two conditions if analysis starts once all cell division has ceased?

The data in Figs. 1-3 do include the 15 minutes immediately after injection. The first 15 minutes after injection is the time when the initial increase in roughness occurs when we inhibit tension, and that effect would be obscured if we removed the initial 15 minutes. We note that the scale of the Y-axis was very different when comparing Figs. 2E and 3C, and that the transient increase in roughness after cell divisions are inhibited is much smaller in magnitude than the increase in roughness caused by reducing tension. To clarify this, we rescaled the Y-axis in Fig. 3C to match the limits shown in Fig. 2E. In addition, we repeated the experiment in Fig. 3, and we confirmed a small, transient increase in boundary roughness when cell divisions were inhibited (new Fig. 3A-C), as well as limited linearization of the ME boundary (new Fig. 3D). We updated the Results section in the manuscript (page 8, paragraph 2, line 3):

"Consistent with model predictions, we found that in control embryos, boundary roughness decreased significantly by $22\pm 6\%$ over 40 minutes ($P < 0.01$, Fig. 3A, C-D and Movie 5). In contrast, inhibiting cell division caused a transient increase in ME boundary roughness, which remained significantly higher than in controls ($P < 0.05$, Fig. 3A-B, C-D and Movie 5)."

3) Does the cell mobility assessment distinguish between actual cell movement and cell shape change? Cell elongation in one direction for example would move the centroid of the cell without the cell actually moving. Indeed at 40 mins in figures 3 A and B the control cells appear to be larger and elongated perpendicularly to the ME boundary compared to the "no division" image. This analysis (in silico and in vivo) should be extended to an assessment of cell shape change and the direction of that change (AP vs DV).

We quantified cell density, both *in vivo* and *in silico* (new Fig. S6). We found that inhibiting cell division resulted in smaller cells and an increase in cell density. The smaller cell areas when cell divisions were inhibited are consistent with an increase in tension (Figs. 6-7), which may promote apical constriction in the ectoderm. However, in our model we found that inhibiting cell divisions reduced cell density, suggesting that the reduction in cell mobility when cell divisions were inhibited does not necessarily result from changes in cell morphology. These new results are now described in the manuscript (page 9, paragraph 2, line 10):

"Inhibiting cell divisions led to an increase in cell density *in vivo* (Fig. S6A-C), but not *in silico* (Fig. S6D-F), suggesting that changes in cell morphology are not responsible for the reduced cell mobility when divisions are inhibited."

4) Is the mobility/cell shape analysis limited to only the ectoderm cells? This should be extended to the mesectoderm cells too. Cells on both sides of the boundary will potentially have an effect on how "rough" it is.

Our previous analysis of cell mobility was limited to ectoderm cells at least one cell diameter away from the ME boundary. To determine if inhibiting cell division affected the mobility of ectoderm cells adjacent to the ME boundary or of mesectoderm cells, we quantified the mean squared displacement (MSD) of ectoderm cells at the ME boundary (new Fig. S4) and mesectoderm cells (new Fig. S5) in controls and in embryos treated with dinaciclib. In controls, both ME boundary-adjacent ectoderm cells and mesectoderm cells displayed lower mobility than ectoderm cells away from the boundary (Figs. 5C-D, S4C-D and S5C-D), suggesting that the refinement of ME boundary linearity may be mainly driven by changes in the mobility of ectoderm cells away from the boundary. Consistent with this, we found that inhibiting cell divisions led to a small reduction in the mobility of boundary-adjacent ectoderm cells (Fig. S4C-F), and no significant effect on the mobility of mesectoderm cells (Fig. S5C-F). Together, these results suggest that local cell movements at the boundary are not the primary determinants of ME boundary linearity. The new data have been described in the Results (page 9, paragraph 2, line1):

"We investigated how cell divisions affected the mobility of cells immediately adjacent to the ME boundary. Both ectoderm cells adjacent to the boundary and mesectoderm cells displayed reduced mobility in controls with respect to ectoderm cells away from the boundary (Figs. 5A-B, S4A-B and S5A-B): the MSDs of cell centroids decreased by 61% in ectoderm cells adjacent to the boundary ($P < 0.05$, Figs. 5D-F and S4D-F), and by 76% in the mesectoderm ($P < 0.01$, Figs. 5D-F and S5D-F). Inhibiting cell divisions had a more modest effect in cells adjacent to the ME boundary as compared with ectoderm cells away from the boundary, with no significant changes in the half time of self-overlap decrease or the MSD when cell divisions were inhibited (Figs. S4C and S5C). These results further suggest that ectoderm cell behaviours away from the ME boundary play an important role in promoting boundary refinement."

5) In figures 5 A and B there is a clear difference in the cell movements (or potentially shape change) seen. The control cell moves mainly in the DV axis, while in the absence of cell divisions this movement appears to be mainly in the AP axis. An assessment of AP vs DV movement should be made as a directional movement/shape change would potentially have an effect on the forces acting on the ME boundary.

Our measurements of MSD are conducted on time-lapse sequences registered to the position of the mesectoderm. This minimizes cell movement along the anterior-posterior axis related to the extension and retraction of the ectoderm (Irvine and Wieschaus, 1994; Schock and Perrimon, 2002). We have clarified this in two different sections:

- Results (page 8, paragraph 3, line 14): "We obtained similar results when we used the mean-squared displacement (MSD) of the cell centroids to quantify cell movements with respect to the mesectoderm: in controls, the MSD of ectoderm cells at 40 min decreased by 83% when cell divisions were inhibited ($P < 0.001$, Fig. 4D-F)."

- Materials and methods (page 17, paragraph 1, line 1): "The MSD was measured in time-lapse sequences registered to the position of the mesectoderm, to minimize cell movements associated with germband elongation and/or retraction which are not included in our model (Irvine and Wieschaus, 1994; Schock and Perrimon, 2002)."

To confirm that we are mainly quantifying movement with respect to the mesectoderm, we separated the AP vs. DV components of cell movement (new Figs. 5E-F, S4E-F and S5E-F). Any effects were predominantly in the DV-oriented cell movements (with respect to the ME boundary).

6) The orientation of cell divisions should be assessed - As they are linked to cell shape they will both be affected by and play a role in determining the direction the tissue grows in. This analysis should be separated into cells in contact with and not contacting the ME boundary to assess for local effects.

We previously conducted these measurements (Yu et al., 2021) (see response to major point #1 above): ectoderm cells adjacent to the boundary divide preferentially with the spindle oriented parallel to the DV axis of the embryo, while ectoderm cells away from the boundary do not have a clear bias in the orientation of their divisions. We have indicated how cell divisions were oriented in the Results (page 5, paragraph 2, line 10):

"Ectoderm cells were stochastically selected to divide, with a frequency and orientation based on experimental data (Yu et al., 2021): cells adjacent to the ME boundary oriented their spindle preferentially along the dorsal-ventral axis, while the rest of the cells divided with random spindle orientations."

7) The analysis of tension distribution should also be carried out in the in vivo movies. Does it match the findings of the in silico analysis? This should be extended to direction of tension relative to AP and DV axis, not just general increase vs decrease. It should also be examined on a more local level - how does tension in the cells immediately either side of ME boundary change compared to the tissue further away?

We used laser ablation to determine whether cell division affects tension in ectoderm cells adjacent to the ME boundary (new Fig. S7, the experiments in Fig. 7 target ectoderm cells far from the boundary). We found that inhibiting cell division did not affect tension in ectoderm cell junctions adjacent to the ME boundary. These results suggest that cell divisions specifically fluidize ectoderm cells away from the ME boundary. We have included the new data in the Results (page 10, paragraph 3, line 8):

"The increase in tension when cell divisions were inhibited was specific to junctions away from the boundary (Fig. S7A-E), and was not associated with changes in the planar polarity or levels of myosin (Fig. S8A-E)."

To investigate how cell division affects the distribution of force, we used the localization of myosin as a proxy for tension (new Fig. S8). As previously reported, myosin was polarized in the ectoderm in control embryos, preferentially localizing to junctions parallel to the dorsal-ventral axis. Blocking cell division did not affect myosin polarization to DV-oriented junctions. These

data suggest that disrupting cells divisions does not alter the distribution of tension in the ectoderm. We discuss the new results in the Results (page 10, paragraph 3, line 8):

"The increase in tension when cell divisions were inhibited was specific to junctions away from the boundary (Fig. S7A-E), and was not associated with changes in the planar polarity or levels of myosin (Fig. S8A-E)."

8) In the discussion it is suggested that divisions prevent the build up of junctional acto- myosin. This should be tested by staining for phosphorylated myosin.

Because the internalization of the mesectoderm occurs over a relatively short period of time, difficult to capture in fixed embryos, and to ensure that we are examining consistent developmental stages, we investigated the effects of cell division on the actomyosin cytoskeleton using a live reporter of myosin localization (myosin:GFP) (new Fig. S8). We quantified junctional GFP levels 30 min after drug treatment, and we found that inhibiting cell division did not affect junctional myosin levels or polarity. We discussed our new data in the Results (page 10, paragraph 3, line 8):

"The increase in tension when cell divisions were inhibited was specific to junctions away from the boundary (Fig. S7A-E), and was not associated with changes in the planar polarity or levels of myosin (Fig. S8A-E)."

We also adjusted the Discussion to reflect that other changes to cytoskeletal dynamics (e.g. turnover) may be important to understand how cell divisions limit tissue tension (page 13, paragraph 1, line 10):

"Thus, cell divisions in the ectoderm may destabilize cell-cell junctions, possibly increasing actomyosin turnover and limiting tension (Fernandez-Gonzalez et al., 2009). Studies quantifying how blocking cell divisions in the ectoderm affects the turnover of adherens junctions and cortical actomyosin may shed light on how cell division facilitates cell rearrangements for boundary refinement."

Minor comments

1) Figure 1 A-D - are these "zoom-ins" of a larger simulation or do the configuration of the cells TO vary? The starting images don't match. Can this be made clear please.

The starting configurations do not match, as they were randomly created (as indicated in the Materials and methods), by generating Voronoi tessellations of randomly-distributed points. We have clarified the protocol to generate the initial cell configurations in the model in the Materials and methods (page 18, paragraph 3, line 1):

"We initialized the simulations with 400 random cells in a periodic box. The initial configurations were generated using Voronoi tessellations of randomly distributed points."

2) About figure 4 - "In our computational model, the degree of self-overlap in the ectoderm in the presence of cell divisions decreased continuously, reaching half of its final value by 31 min - I don't understand this? At 31min it's at about 0.85 and the final value is about 0.75 - which isn't half. Likewise for the statement about the treated sample too. This line doesn't drop at all - there is no half-time. Please can this be clarified.

We were referring to the time to reach half of the change in value. In the simulations in which cell divisions were inhibited, we only measured a "half-time" of self-overlap reduction for the 14/80 simulations that displayed a decrease. We have rephrased the discussion of changes in self-overlap in the model (page 8, paragraph 3, line 7):

"In our computational model, the degree of self-overlap in the ectoderm in the presence of cell

divisions decreased continuously, with half of the reduction (the half-time of self-overlap decrease) occurring by 31.1 ± 0.4 min ($P < 0.001$, Fig. 4A, C). In contrast, when we inhibited cell divisions in the model, the self-overlap function did not decrease for 83% of simulations (66/80), and for the remaining simulations (14/80) the half-time of self-overlap reduction was 37.7 ± 0.5 min (Fig. 4B-C), significantly longer than control simulations ($P < 0.001$), suggesting that the cells were frozen when cell divisions were inhibited."

3) Figure 6 A at 40 mins - there are grey cells in the mesectoderm - does this mean they under compression? Please clarify

The gray junctions in the simulations (Fig. 6A-B) are compressed, consistent with the progressive internalization of the mesectoderm. We have clarified this point in the figure legend (page 6, paragraph 1, line 3):

"Gray junctions are compressed."

4) In figure 6 there are also differences in the mesectoderm, these should also be discussed

We added a sentence to the Discussion describing the changes in mesectoderm tension in the model when cell divisions in the ectoderm were inhibited (page 12, paragraph 2, line 13):

"Our model also predicted that mesectoderm cell-cell contacts sustain compressive stress in controls. In contrast, when ectoderm divisions were inhibited, mesectoderm junctions predominantly sustained tensile stress, consistent with our data that ectoderm divisions push against the ME boundary, and suggesting that polarized myosin cables in the ectoderm may pull on the ME boundary, locally increasing boundary roughness and resisting mesectoderm internalization."

5) Figure 7. Where were the ablation experiments done - were they on boundary cells or not? Were the junctions cut always AP or DV oriented etc... Please specify.

Please, see response to major point #7 above. Our original ablation experiments were conducted in ectoderm cells two or three cell rows away from the ME boundary. We targeted cell-cell junctions parallel to the DV axis of the embryo, as the shorter AP-oriented junctions are harder to sever without affecting additional vertices and/or cell-cell contacts. We have now used laser ablation to determine whether cell divisions affect tension in ectoderm cells adjacent to the ME boundary (new Fig. S7), and we have investigated the distribution of myosin to examine changes in the axis of tension orientation (new Fig. S8). We found that in cells immediately adjacent to the boundary, there was no significant increase in tension after dinaciclib treatment. In addition, our measurements indicate that the polarized distribution of myosin in the ectoderm is not affected by inhibiting cell division. Thus, our results indicate that cell divisions away from the ME boundary modulate ectoderm tension (but not the distribution thereof) to facilitate boundary refinement. We have included the new data in the Results (page 10, paragraph 3, line 8):

"The increase in tension when cell divisions were inhibited was specific to junctions away from the boundary (Fig. S7A-E), and was not associated with changes in the planar polarity or levels of myosin (Fig. S8A-E)."

6) Figure 5 A and B - can a colour coding for time be added to the tracks please.

To facilitate interpretation of the different groups in the figures, instead of colours, we have added beginning (triangle) and end (cross) markers for each one of the cell trajectories displayed in Figs. 4, 5, S5 and S6. This has been indicated in the figure legends:

"Red triangles and crosses indicate the starting and final points of the trajectories, respectively."

Reviewer 2: The authors study the mesectoderm-ectoderm boundary of the early *Drosophila* embryo as a model system for tissue boundary maintenance. In line with previous findings, they

show that acto-myosin generated tension stabilizes the boundary against fluctuations generated by cell divisions. Remarkably they find that in the presence of acto-myosin generated tension, cell divisions can refine (or sharpen) the boundary.

Overall, the manuscript is well written and will be of interest to the readership of Development. However, before I can recommend the manuscript for publication, the authors should address the following points.

1. The main claim is that cell divisions help increase tissue fluidity and thereby refine the tissue boundary. In the vertex model, there are several ways to increase tissue fluidity. In addition to increasing the cell division rate, the target shape index or the "temperature" (strength of force fluctuations), could be increased to drive a solid-to-fluid transition. These two variants should be tested in the model to demonstrate that it is indeed the increase in fluidity that helps sharpen the tissue boundary.

We examined the effect that changes to the target shape index and Brownian noise have on cell mobility *in silico*. Briefly, we increased the target shape index (p_0), and we found that, as expected, cell movements increased (Fig. R1A-B, E). The increased mobility of cells with a greater target shape index partially rescued cell mobility when cell divisions were inhibited (Fig. R1C-D) and limited the secondary increase in roughness (Fig. R1E). We found similar results when we increased Brownian noise (Fig. R2A-E). Together, these results suggest that ectoderm cell movement (regardless of the mechanism) fluidizes the tissue in a process key for ME boundary refinement.

In our experiments, we maintained constant values of target shape index and "temperature", as we are investigating the role of cell divisions in boundary maintenance. Other model parameters, such as friction, would also likely affect tissue fluidity. Therefore, we believe that the requested *in silico* experiments could be combined with additional biological tests (e.g. increasing Brownian noise possibly by manipulating environmental temperature, or altering cell-cell or cell-matrix adhesion), which is the focus of work of other trainees in our labs. Thus, we decided not to include the results of the simulations with increased target shape index or Brownian noise in the manuscript, and we will follow up in a future study.

We added a note clarifying that multiple factors can affect tissue fluidity, and that cell division is only one of them, in the Discussion (page 13, paragraph 1, line 14):

"It is important to note that tissue fluidity could increase through multiple mechanisms, including changes in cell-cell and cell-substrate interactions (friction) (Founounou et al., 2021; Staddon et al., 2022), Brownian noise (temperature) (Bi et al., 2016; Devany et al., 2021), or cell elongation (target cell shape) (Bi et al., 2015). Cell divisions perpendicular to the plane of the tissue could also fluidize tissues. Future work should extend and take advantage of vertex models to determine how different mechanisms that control tissue fluidity affect boundaries, to establish the uniqueness of the role of cell division on boundary maintenance."

Figure R1. Cell elongation contributes to tissue fluidity and boundary refinement *in silico*. (A-D) Simulations of mesectoderm ingressions in tissues with control target shape index (A, C, $q = 3.4$) or with an increased target shape index (B, D, $q = 3.7$), with (A-B) or without (C-D) cell divisions. Magenta, ectoderm; teal, mesectoderm. Bars, 20 μm . Anterior, left. Time zero corresponds to the time in which the mesectoderm width starts decreasing. (E-F) Self-overlap (E) and relative boundary roughness (F) in control simulations (blue, $n = 40$ simulations), with increased q (purple, $n = 40$), with no divisions (green, $n = 40$), or with increased q and no divisions (cyan, $n = 40$). Error bars, s.e.m..

Figure R2. Brownian noise contributes to tissue fluidity and boundary refinement *in silico*. (A-D) Simulations of mesectoderm ingression in tissues with control levels of Brownian noise (A, C, $T = 0.010$) or with increased Brownian noise (B, D, $T = 0.015$), with (A-B) or without (C-D) cell divisions. Magenta, ectoderm; teal, mesectoderm. Bars, 20 μm . Anterior, left. Time zero corresponds to the time in which the mesectoderm width starts decreasing. (E-F) Self-overlap (E) and relative boundary roughness (F) in control simulations (blue, $n = 40$ simulations), with increased T (purple, $n = 40$), with no divisions (green, $n = 40$), or with increased T and no divisions (cyan, $n = 40$). Error bars, s.e.m..

2. In the first paragraph of the results section the authors write "Ectoderm cells were randomly selected to divide, with a frequency and orientation based on experimental data (Yu et al., 2021)." It would be good if the authors would briefly mention what this frequency and orientation are to make the manuscript more self-contained. Moreover, the authors could test other division orientations in the model. Would divisions perpendicular to the boundary challenge but not refine it?

Please, see response to Reviewer 1, major point #1. Briefly, we have now included the information regarding cell division orientation at the beginning of the Results (page 5, paragraph 2, line 10):

"Ectoderm cells were stochastically selected to divide, with a frequency and orientation based on experimental data (Yu et al., 2021): cells adjacent to the ME boundary oriented their spindle preferentially along the dorsal-ventral axis, while the rest of the cells divided with random spindle orientations."

In addition, we investigated if changes in cell division orientation in the model affected boundary alignment (new Fig. S1). Our previous results showed that ectoderm cell divisions challenge the boundary (Fig. 2). In the ectoderm, only the divisions of cells adjacent to the boundary are oriented, with the spindle parallel to the DV axis of the embryo, perpendicular to the boundary. Thus, we tested whether randomizing the orientation of boundary-adjacent cell divisions, or orienting the divisions parallel to the boundary would facilitate boundary formation. We found that altering the orientation of ectoderm cell divisions adjacent to the boundary did not have a significant impact on the ME boundary. These data suggest that cell division orientation adjacent to the boundary is not a major challenge to the ME interface, consistent with the relatively low number of ectoderm cells that divide adjacent to the boundary. These data are now discussed in the Results (page 6, paragraph 2, line 1):

"To further establish how ectoderm cell divisions may impact the ME boundary, we simulated scenarios altering the orientation of cell divisions adjacent to the boundary. We found that randomizing the angle of division of ectoderm cells adjacent to the boundary (Fig. S1A-B, D-E, G), or rotating the angle by 90 degrees, such that the spindle was oriented preferentially along the anterior-posterior axis (Fig. S1A, C-D, F-G) did not affect the increase in roughness when tension at the boundary decreased, or the rescue of roughness when cell divisions were inhibited, consistent with the relatively small number of ectoderm cells adjacent to the boundary."

3. Experimentally, the authors could test whether changing the division axis, i.e. to become vertical (cleavage plane perpendicular to the apico-basal axis), changes the effect of cell divisions on the tissue boundary. One might hypothesize that vertical divisions would not challenge the boundary but might still help refine it because they might reduce tension in the ectoderm.

Ectoderm and mesectoderm cells divide exclusively on the plane of the tissue (Yu et al., 2021). In addition, our vertex model is two-dimensional. Thus we did not consider cell divisions out of the plane of the tissue. We are currently working on a three-dimensional version of the model in which we will be able to examine the impact of cell divisions perpendicular to the tissue plane on boundary maintenance in other biological systems.

We have included a note in the Discussion about how additional mechanisms could control tissue fluidity (page 13, paragraph 1, line 14):

"It is important to note that tissue fluidity could increase through multiple mechanisms, including changes in cell-cell and cell-substrate interactions (friction) (Founounou et al., 2021; Staddon et al., 2022), Brownian noise (temperature) (Bi et al., 2016; Devany et al., 2021), or cell elongation (target cell shape) (Bi et al., 2015). Cell divisions perpendicular to the plane of the tissue could also fluidize tissues. Future work should extend and take advantage of vertex models to determine how different mechanisms that control tissue fluidity affect boundaries, to establish the uniqueness of the role of cell division on boundary maintenance."

4. In the discussion, the authors state "the origin of that tensile force is not known." referring to the force acting along the mesectoderm-ectoderm (ME) tissue boundary. I don't believe this statement is fair in the light of the extensive literature on the tissue mechanics of germ band extension. Active cell intercalations in the ectoderm drive extension of the germ band and thereby stretch the ME boundary. The authors should mention this key contribution to stretching

of the ME boundary.

Our Discussion already mentions that ectoderm (germband) cells intercalate to dissipate anisotropic stress caused by posterior midgut invagination. We have further extended the Discussion to mention a potential role of cell intercalation in generating anisotropic stress (page 11, paragraph 3, line 3):

"Posterior pulling forces, generated by the invagination of the posterior midgut could generate anisotropic stress along the anterior-posterior axis, dissipated by ectoderm cells through intercalation (Collinet et al., 2015; Lye et al., 2015), but not by mesectoderm cells, which do not exchange neighbours (Wang et al., 2017). This model would thus predict that mesectoderm cells would be the main contributors to the accumulation of myosin at the ME boundary, something that has not been tested. Intercalary cell behaviours contribute to the convergent extension of the ectoderm (Irvine and Wieschaus, 1994; Bertet et al., 2004; Zallen and Zallen, 2004; Blankenship et al., 2006), which generates further anisotropic stress (Rauzi et al., 2008; Fernandez-Gonzalez et al., 2009) that may contribute to the localization of myosin at the ME boundary."

5. In a similar vein, I don't think that the statement "The mechanisms by which cell divisions reduce tissue tension are unclear." is correct. The role of cell divisions on tissue mechanics has been studied extensively. The main effect is that divisions allow the tissue to expand and therefore relax extensile tissue strain. Assuming that the junctions have some (effective) elastic properties, like the area and perimeter elasticity of the vertex model, relaxation of strain implies relaxation of stress. This should be mentioned by the authors.

This statement was referring specifically to the ectoderm in the *Drosophila* embryo, so we have toned it down. In addition, we have added a discussion of how oriented cell divisions adjacent to the ME boundary could contribute to strain and stress relaxation (page 13, paragraph 2, line 13):

"The mechanisms by which cell divisions reduce tissue tension in the ectoderm are unclear. Anisotropic strain can orient the mitotic spindle parallel to the axis of maximum deformation, in a process that releases strain (Campinho et al., 2013; Wyatt et al., 2015). However, ectoderm cells divide with random orientations, except those immediately adjacent to the ME boundary, which divide with their spindle perpendicular to the boundary (Yu et al., 2021). Thus, it is possible that mesectoderm internalization generates a local, dorsal-ventral-oriented pulling force that orients the divisions of ectoderm cells adjacent to the ectoderm, relaxing strain and tension throughout the tissue."

Minor points:

1. The authors should justify their choice of model parameters, in particular the target shape index which plays a key role for the material properties of the vertex model tissue. The "temperature" should be called noise strength to avoid confusion with the actual temperature.

The choice of target shape index was based on our previous work (Yu et al., 2021), with the goal of recapitulating the heterogeneity in cell areas observed *in vivo* for the two different cell types considered (ectoderm and mesectoderm). We have indicated this in the Materials and methods (page 19, paragraph 1, line 3):

"We used a constant value of $q = 3.4$ as the target shape index to scale the preferred perimeter for all cell types while allowing the cell type-specific cell size heterogeneity observed *in vivo*."

We have explained the meaning of the temperature parameter in the Materials and methods (page 18, paragraph 2, line 6):

"The temperature T represents the strength of the Brownian noise in the position of cell vertices (Brańka and Heyes, 1999)."

2. The measure of boundary roughness Eq. (2) could be justified better and made more systematic. Why is it measured along 13 μm segments? Are the results robust when changing the segment length? A more systematic way to measure the roughness would be tortuosity, commonly defined as the arclength divided by the end-to-end distance of a curve.

We modified our roughness measurement to be simply the standard deviation of the Y-coordinates after rotating the boundary to be horizontal and detrending the Y-coordinates so that their mean is zero. This avoids the need to use a segment length parameter, and leads to similar results as the previous measurement.

We have applied the new roughness metric throughout the document (e.g. new Figs. 1, 2, 3) and defined the metric in the Materials and methods (page 16, paragraph 2, line 1):

"To measure the linearity of the ME boundary, we annotated the interface between ectoderm and mesectoderm using the LiveWire algorithm. The LiveWire annotation was rotated so that a fit line was horizontal, and detrended so that the mean Y-coordinate was zero. We quantified the roughness of the boundary as the standard deviation of the Y-coordinates of the pixels on the boundary after rotation and detrending."

3. Typos:

In Eq. (8) it should read $P_i - P_{0i}$, instead of $P_i + P_{0i}$

Fixed.

Reviewer 3: The paper establishes the mesoderm/ectoderm compartment boundary in the early fly embryo as a new experimental system for the quantitative analysis of the mechanisms by which developing organisms control borders between different cell and tissue types. The main observation is that cell divisions can both disrupt the straightness of the border and relieve tissue tension. These results are based on quantitative analysis of imaging data, laser ablation experiments, and pharmacological perturbations. The results are also supported by and, to a significant extent, supported by a clear model, based on the vertex description of the epithelium with an appropriate modification for region-specific control of tissue tensions. The manuscript was a pleasure to read. The authors did an excellent job motivating their work and presented results in a very logical way. I see no flaws in the study and am happy to recommend it for publication.

SUGGESTIONS TO AUTHORS

1. A couple of suggestions: it would be good to discuss the robustness of the model fits and try to come up with a dimensionless group that can be used to classify different types of effects of cell division on boundary straightness.

Please, see responses to Reviewer 1, major point #1. Briefly, we examined how changes in the orientation of cell division affect boundary roughness (new Fig. S1). Ectoderm cells adjacent to the boundary orient their spindle perpendicular to the ME interface (Yu et al., 2021), which is the orientation that we originally used in our modeled. Thus, we tested whether randomizing the orientation of boundary-adjacent cell divisions, or orienting the divisions parallel to the boundary would alter boundary roughness. We found that changing the orientation of ectoderm cell divisions adjacent to the boundary did not have a significant impact on the roughness of the ME boundary. These data suggest that cell division orientation adjacent to the boundary is not a major challenge to the ME interface, consistent with the relatively low number of ectoderm cells that divide adjacent to the boundary. These data are now discussed in the Results (page 6, paragraph 2, line 1):

"To further establish how ectoderm cell divisions may impact the ME boundary, we simulated scenarios altering the orientation of cell divisions adjacent to the boundary. We found that randomizing the angle of division of ectoderm cells adjacent to the boundary (Fig. S1A-B, D-E, G), or rotating the angle by 90 degrees, such that the spindle was oriented preferentially along the anterior-posterior axis (Fig. S1A, C-D, F-G) did not affect the increase in roughness when tension at the boundary decreased, or the rescue of roughness when cell divisions were inhibited, consistent with the relatively small number of ectoderm cells adjacent to the boundary."

We also established the effect of inhibiting cell divisions *in silico* in ectoderm cells immediately adjacent to the boundary or away from the boundary when tension decreased (new Fig. S2). We found that cell divisions away from the boundary had the greatest impact on boundary roughness when tension was disrupted, probably because the number of away cell divisions is significantly greater than the number of boundary- adjacent cell divisions. These new data are now discussed in the Results (page 6, paragraph 2, line 7):

"Instead, the rescue in boundary roughness in the absence of tension could only be accomplished when all cell divisions (Fig. S2A-B, F) or cell divisions far from the boundary (Fig. S2C-F) were inhibited or reduced in number, suggesting that the number of ectoderm cell divisions, rather than their position or orientation, challenges the ME interface."

2. I would also like to see a more clear discussion of the differences of the general effects in this study and the work on compartment boundaries in the wing imaginal disc, especially since that system was used to establish a lot of ideas and tools used by the authors.

We have discussed what we consider to be the main difference between the ME boundary and other tissue-tissue boundaries, including compartment boundaries in the wing imaginal disc: the ME boundary increases its linearity despite a continuous reduction in myosin levels. This difference is now highlighted in the Introduction, and provides a rationale to further investigate the mechanisms of boundary maintenance beyond actomyosin contractility (page 4, paragraph 3, line 6):

"Importantly, and in contrast to compartment boundaries in the wing disc or the embryo (Major and Irvine, 2005; Monier et al., 2010; Röper, 2012), the actomyosin cable at the ME boundary is disassembled over time, in a process thought to contribute to the internalization of the mesectoderm (Yu et al., 2021). Despite the reduction in myosin levels, the ME boundary remains linear. The mechanisms that maintain boundary linearity while myosin levels decrease are not understood."

Interestingly, it has been proposed that a mechanism independent of actomyosin-based contractility must maintain the DV boundary at late stages of *Drosophila* larval development (Monier et al., 2011), when an actin cable is not present at the boundary. We have discussed how cell divisions could be this alternative mechanism (page 13, paragraph 2, line 2):

"Surprisingly, ectoderm divisions also refine the boundary. Thus, proliferating epithelia may maintain pre-established boundaries when contractile cables are disassembled, for instance at the dorsal-ventral boundary in the wing disc of late *Drosophila* larvae (Monier et al., 2011). The mechanisms that establish and maintain tissue boundaries are conserved (Sánchez-Corrales and Röper, 2018). Thus, our findings may reveal a conserved mechanism whereby boundaries across proliferative tissues not only resist, but also benefit from cell divisions."

3. Finally, it would be great to have a more resolved view of local tissue geometry changes accompanying a mitotic event.

Please, see response to Reviewer 1, major point #3. Briefly, in our computational model cells do not change shape as they divide, and thus, we have not considered cell shape changes during cell division *in vivo*. To assess changes in local tissue geometry, we measured changes in cell density (new Fig. S6). While inhibiting cell divisions *in vivo* increased cell density (possibly due to the

increase in tension driving apical constriction), inhibiting cell divisions *in silico* had the opposite effect and reduced cell density, suggesting that the impact of cell divisions on cell mobility and boundary roughness is independent of cell morphology changes. We have included these results in the Results (page 9, paragraph 2, line 10):

"Inhibiting cell divisions led to an increase in cell density *in vivo* (Fig. S6A-C), but not *in silico* (Fig. S6D-F), suggesting that changes in cell morphology are not responsible for the reduced cell mobility when divisions are inhibited."

Second decision letter

MS ID#: dev.204817R1

MS TITLE: Cell divisions both challenge and refine tissue boundaries in the Drosophila embryo

AUTHORS: Veronica Castle, Merdeka Miles, Rafael Perez-Vicente, Rodrigo Fernandez-Gonzalez and Gonca Erdemci-Tandogan

Dear Dr Fernandez-Gonzalez,

I am happy to tell you that your manuscript has been accepted for publication in Development, pending our standard publication integrity checks.

Reviewer 1

Advance summary and potential significance to field

This work shows that, as previously known at other tissue boundaries, that cell divisions challenge the ME boundary. However, more surprisingly it all also demonstrates that divisions also help to define/develop the ME boundary by increasing the fluidity of the tissue (by releasing tissue tension). It's argued that this allows greater cellular movement, which in turn reduces the roughness of the boundary. This beneficial role of cell division in defining tissue boundaries is an interesting development in the field of tissue biology/mechanics.

Comments for the author

The authors have very comprehensively addressed my previous comments and I am very happy to recommend this manuscript for publication.

As noted in the discussion I too am interested in how division releases tissue tension, particularly given that inhibiting division doesn't noticeably alter myosin levels or distribution? Is it somehow linked to the smaller cell sizes/increase in density?

Likewise, as it is clear this affect is limited to the non-boundary ectoderm cells, it would be interesting to investigate how it is "communicated" to the boundary cells - how are they coupled?

Reviewer 2

Advance summary and potential significance to field

The authors have addressed all my comments. They deferred some of the additional simulations I asked for to a future publication, which is ok with me. I recommend the manuscript for publication in its current form.

In the response to the comments by reviewer 1 I noticed that the reply to the reviewer's point 3 does not appear to answer the question. The question is on cell motility and shape while the answer is about density. I don't think this is a major issue though, so it shouldn't stand in the way of publication.